# Cost-Sensitive Learning to Defer to Multiple Experts with Workload Constraints

**Jean V. Alves**                                         *jean.alves@feedzai.com*
*Feedzai*

**Diogo Leitão**
*Feedzai*

**Sérgio Jesus**                                           *sergio.jesus@feedzai.com*
*Feedzai*

**Marco O.P Sampaio**
*Feedzai*

**Javier Liébana**                                         *javier.liebana@feedzai.com*
*Feedzai*

**Pedro Saleiro**                                          *pedro.saleiro@feedzai.com*
*Feedzai*

**Mário A. T. Figueiredo**                                 *mario.figueiredo@tecnico.ulisboa.pt*
*Instituto Superior Técnico, ULisboa*
*Instituto de Telecomunicações*

**Pedro Bizarro**                                          *pedro.bizarro@feedzai.com*
*Feedzai*

**Reviewed on OpenReview:** *https://openreview.net/forum?id=TAvGZm2Rqb*

## Abstract

*Learning to defer* (L2D) aims to improve human-AI collaboration systems by learning how to defer decisions to humans when they are more likely to be correct than an ML classifier. Existing research in L2D overlooks key real-world aspects that impede its practical adoption, namely: i) neglecting cost-sensitive scenarios, where type I and type II errors have different costs; ii) requiring concurrent human predictions for every instance of the training dataset; and iii) not dealing with human work-capacity constraints. To address these issues, we propose the *deferral under cost and capacity constraints framework* (DeCCaF). DeCCaF is a novel L2D approach, employing supervised learning to model the probability of human error under less restrictive data requirements (only one expert prediction per instance) and using constraint programming to globally minimize the error cost, subject to workload limitations. We test DeCCaF in a series of cost-sensitive fraud detection scenarios with different teams of 9 synthetic fraud analysts, with individual work-capacity constraints. The results demonstrate that our approach performs significantly better than the baselines in a wide array of scenarios, achieving an average 8.4% reduction in the misclassification cost. The code used for the experiments is available at `https://github.com/feedzai/deccaf`

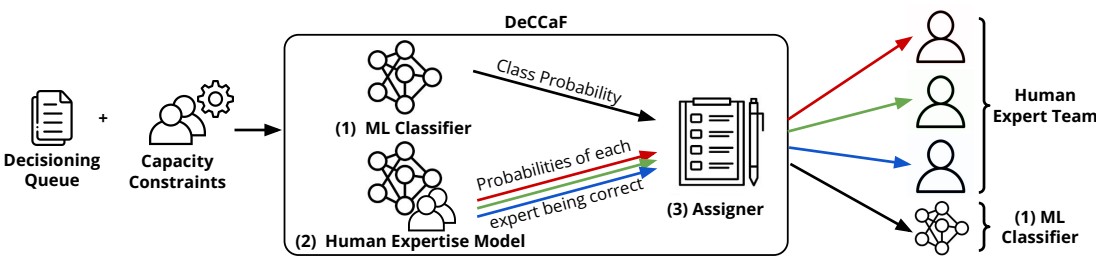

Figure 1: Schematic Representation of DeCCaF

# 1   Introduction

An increasing body of recent research has been dedicated to human-AI collaboration (HAIC), with several authors arguing that humans have complementary sets of strengths and weaknesses to those of AI (De-Arteaga et al., 2020; Dellermann et al., 2019). Collaborative systems have demonstrated that humans are able to rectify model predictions in specific instances (De-Arteaga et al., 2020), and have shown that humans collaborating with an ML model may achieve synergistic performance: a higher performance than humans or models alone (Inkpen et al., 2022). In high-stakes scenarios where ML models can outperform humans, such as healthcare (Gulshan et al., 2016), HAIC systems can help address safety concerns (*e.g.*, the effect of changes in the data distribution (Gama et al., 2014)), by ensuring the involvement of humans in the decision-making process.

The state-of-the-art framework to manage assignments in HAIC is *learning to defer* (L2D) (Charusaie et al., 2022; Hemmer et al., 2022; Raghu et al., 2019b;a; Mozannar & Sontag, 2020b; Mozannar et al., 2023; Madras et al., 2018b; Steege, 2023; Verma & Nalisnick, 2022a; Verma et al., 2023). L2D aims to improve upon previous approaches, such as rejection learning (Chow, 1970; Cortes et al., 2016), which defer based solely on the ML model's confidence, by also estimating the human confidence in a given prediction and passing the instance to the decision-maker who is most likely to make the correct decision.

Previous work in L2D does not address several key aspects of collaborative systems. In real-world scenarios, multiple human experts are employed to carry out the classification task, as the volume of instances to process cannot be handled by a single human. However, only a small subset of L2D research focuses on the multi-expert setting, where decisions have to be distributed throughout a team comprised of a single ML classifier and multiple human experts (Keswani et al., 2021; Hemmer et al., 2022; Verma et al., 2023). To the best of our knowledge, Verma et al. (2023) propose the only two consistent multi-expert L2D formulations, with both approaches assuming the existence of every expert's predictions for all training instances. In real-world applications, we often have more limited data availability, with only a single expert (De-Arteaga et al., 2020) or a small subset of the experts (Gulshan et al., 2016) providing predictions for each instance. This means that a practitioner aiming to train L2D algorithms would have to purposefully collect the set of every expert's predictions for a sample of instances, which could incur large costs or even be unfeasible. We will propose L2D architectures that allow for assigners to be trained assuming that each instance is accompanied by the prediction of only one expert out of the team.

Current multi-expert L2D methods also neglect human capacity limitations. Should there be an expert that consistently outperforms the model and all the other experts, the optimal assignment would be to defer all cases to that expert, which in practice is not feasible. Furthermore, current work neglects cost-sensitive scenarios, where the cost of misclassification can be class or even instance-dependent (*e.g.*, in medicine, false alarms are generally considered less harmful than failing to diagnose a disease).

To address the aforementioned L2D limitations, we propose the *deferral under cost and capacity constraints framework* (DeCCaF): a novel deferral approach to manage assignments in cost-sensitive human-AI decision-making, which respects human capacity constraints. Our method is comprised of three separate components, represented schematically in Figure 1: (1) an ML classifier modelling the probability of the target class given the instance features; (2) a human expertise model (HEM) that models the probabilities of correctness of

each of the experts in the team; and (3) an assigner that computes the best possible set of assignments given misclassification costs and capacity constraints.

Due to the lack of sizeable datasets with multiple human predictions and to the high costs associated with producing one, we empirically validate DeCCaF in a series of realistic cost-sensitive fraud detection scenarios, where a team of 9 synthetic fraud analysts and one ML classifier are tasked with reviewing financial fraud alerts. We conclude that, across all scenarios, DeCCaF performs similarly to, or significantly better than L2D baselines, as measured by average misclassification costs.

In summary, our contributions are:

- DeCCaF: a novel L2D method that models human behavior under limited data availability, using constraint programming to obtain the optimal set of assignments (Section 3).

- A novel benchmark of complex, feature-dependent synthetic expert decisions, in a realistic financial fraud detection scenario (Section 4.1).

- Experimental evidence that our approach outperforms baselines in a set of realistic, cost-sensitive fraud detection scenarios (Section 5).

In Section 2, we describe the relevant related work, focusing on recent developments in multi-expert L2D and examining the shortcomings of the deferral systems proposed so far. We also discuss current practices in L2D evaluation, particularly the use of synthetically generated expert predictions, and how they can be improved. We then describe DeCCaF in Section 3, first by formulating a novel training method, compatible with limited data availability, for the commonly used classifier-rejector framework (Mozannar & Sontag, 2020a; Verma & Nalisnick, 2022a; Verma et al., 2023; Cortes et al., 2016), demonstrating that it produces optimal unconstrained deferral. As the traditional classifier-rejector approach fails to consider the existence of human work-capacity limitations, we propose a novel formulation for global assignment optimization under capacity constraints. In Section 4 we detail a realistic fraud detection experimental setup as well as the method for the synthetic expert prediction generation. We also discuss the capacity-aware baselines used in our L2D benchmark. The experimental results are reported and analyzed in Section 5.

## 2 Related Work

### 2.1 Current L2D Methods

The simplest deferral approach in the literature is *rejection learning* (ReL), which dates back to the work of Chow (1970); Cortes et al. (2016). In a HAIC setting, ReL defers to humans the instances that the model rejects to predict (Madras et al., 2018b; Raghu et al., 2019a). A simple example (Hendrycks & Gimpel, 2017) is to obtain uncertainty estimates of the model prediction for each instance, rejecting to predict if the uncertainty estimate is above a given threshold.

Madras et al. (2018b) criticize ReL, arguing that it does not consider the performance of the human involved in the task and propose *learning to defer* (L2D), where the classifier and assignment system are jointly trained, taking into account a single model and a single human and accounting for human error in the training loss. Many authors have since contributed to the single-expert framework: Mozannar & Sontag (2020b) show that the loss proposed by Madras et al. (2018b) is inconsistent, proposing a consistent surrogate loss that yields better results in testing; Verma & Nalisnick (2022b) critique the approach of Mozannar & Sontag (2020b), demonstrating that their surrogate loss has a degenerate parameterization, causing miscalibration of the estimated probability of expert correctness. Keswani et al. (2021) observe that decisions can often be deferred to one or more humans out of a team, expanding L2D to the multi-expert setting. Verma et al. (2023) propose the first consistent and calibrated surrogate losses for the multi-expert setting, by adapting the work of Verma & Nalisnick (2022b) and the softmax surrogate loss of Mozannar & Sontag (2020b). All aforementioned studies focus on deriving surrogates for the $0-1$ loss, meaning they are not directly applicable to cost-sensitive scenarios, where the cost of erring can be class-dependent (i.e., different costs

for false positive and false negative errors), or even instance-specific (i.e., every instance has an associated misclassification cost).

Another key facet of L2D research is the interplay between the two components of an assignment system: the rejector (which decides whether to defer and to whom) and the classifier (which produces automatic predictions if the rejector chooses not to defer). Mozannar & Sontag (2020b) argue that the main classifier should specialize on the instances that will be assigned to it, in detriment of those that will not. This is done by jointly training the classifier and the rejector, without penalizing the classifier's mistakes on instances that the rejector defers to an expert. The approach of Verma et al. (2023) differs in that the rejector and the classifier are trained independently, meaning that the classifier is encouraged to predict correctly on all instances, likely to be deferred or not. However, in the multi-expert setting, the two-stage one-vs.-all approach of Verma et al. (2023) outperforms their adaptation of the softmax loss of Mozannar & Sontag (2020a), which employs joint-learning, due to the same calibration problems observed in the single-expert setting. A more recent joint learning approach proposed by Mozannar et al. (2023) is theoretically shown to enable the system to achieve an improvement in overall performance, as their surrogate loss differs from previous work in that it is realizable $\mathcal{H}$-consistent. In more recent work, however, Mao et al. (2024) demonstrate that two-staged learning, the separate training of the classifier and rejector, with their proposed surrogate losses, can also guarantee realizable consistency. We argue that joint learning is not be suitable for real-world applications, as, by design, the instances that are most likely to be deferred are those in which the classifier will perform worse, which would make the system highly susceptible to changes in human availability, should the AI have to predict on instances that were originally meant to be deferred.

As previously mentioned, the key barrier to the adoption of current L2D methods is that they require predictions from every human in the team, for every training instance. Current L2D research tackling data-efficient learning focuses only on the single-expert setting, using active-learning principles (Charusaie et al., 2022) to construct a smaller dataset with every human's predictions for all instances, or employing semi-supervised learning (Hemmer et al., 2023) to impute the missing human predictions.

When deferring, constraints in human work-capacity are rarely considered in multi-expert L2D, where the goal is usually to find the best decision-maker for each instance, disregarding the amount of instances that are deferred to each individual agent. To the best of our knowledge, only Mao et al. (2024) consider control of the amount of deferrals in multi-expert L2D, by allowing for the definition of a constant deferral cost $c_j$ for each available expert $j$, which is included in their realizable $\mathcal{H}$-consistent two stage losses. Similar approaches had been considered in the single-expert case, allowing for the inclusion of a regularization term or expert deferral cost (Mozannar & Sontag, 2020b; Steege, 2023; Narasimhan et al., 2022). While such a formulation allows for regularization of deferral across experts, it would not be adequate for a real-world system, where the availability of experts may vary drastically over time. In a real-world system, different teams of experts may work at different schedules or have days off, which would result in experts being available during certain time intervals while absent in others.

The existence of changes in expert availability within real-world HAIC scenarios means that an L2D system must be able to accommodate expert capacity constraints which are defined at inference-time, and not during training. This restriction is similar to that considered by Zhou et al. (2022) in the context of sparsely activated mixture-of-experts models, where the routing strategy (*i.e.*, how tokens are distributed throughout experts) is akin to the task of L2D algorithms. The authors propose allowing for a hard constraint on the number of tokens selected by each expert and the number of experts each token is routed to in inference. This approach allows for dynamic limitation of the compute resources necessary, without needing to retrain their models.

In our work, we propose a multi-expert L2D algorithm that can be used in cost-sensitive scenarios, trained with restrictive data requirements, and taking into account individual human work-capacity constraints that are imposed only at inference time.

## 2.2 Simulation of Human Experts

Due to the lack of sizeable, public, real-world datasets with multiple experts, most authors use *label noise* to produce arbitrarily accurate expert predictions on top of established datasets found in the ML literature.

Mozannar & Sontag (2020b) use CIFAR-10 (Krizhevsky et al., 2009) and simulate an expert with perfect accuracy on a fraction of the 10 classes, but random accuracy on the others (see also the work by Verma & Nalisnick (2022b) and Charusaie et al. (2022)). The main drawback of these synthetic experts is that their expertise is rather simplistic, being either feature-independent or only dependent on a single feature or concept. This type of approach has been criticised (Zhu et al., 2021; Berthon et al., 2021), and *instance-dependent label noise* (IDN) has been proposed as a more realistic alternative, as human errors are likely to be dependent on the difficulty of a given task, and, as such, should depend on its features. In this work, we propose an IDN approach to simulate more complex and realistic synthetic experts.

## 3 Method

The main goal of our work is to develop a multi-expert assignment system that is able to optimize assignments in cost-sensitive tasks, subject to human work-capacity constraints. For this method to be applicable in real-world scenarios, it is crucial that the assigner can be trained with limited human prediction data (only one expert prediction per instance) and that, in inference, our method be robust to variations in expert work-capacity.

To tackle these objectives, we propose DeCCaF: a novel assignment system that optimizes instance allocation to a team of one or more analysts, while respecting their work-capacity constraints. Given a set of instances with one associated expert prediction, we train a *human expertise model* (HEM) that jointly models the human team's behavior. This model predicts the probability that deferral to a given expert will result in a correct decision. An ML classifier is trained over the same sample with the aim of estimating the likelihood of correctness in automatically predicting on a given instance. We then employ constraint programming (CP) to maximize the global probability of obtaining correct decisions under workload constraints.

### 3.1 Deferral Formulation

**Data** Assume that, for each instance $i$, there is a ground truth label $y_i \in \mathcal{Y}$, a vector $\boldsymbol{x}_i \in \mathcal{X}$ representing its features, a specific cost of misclassification $c_i$, a prediction $m_{j,i} \in \mathcal{Y}$ from a given expert $j \in \{1, ..., J\}$, and $j_i$, identifying which expert made the prediction. The training set can then be represented as $S = \{\boldsymbol{x}_i, y_i, m_{j,i}, c_i, j_i\}_{i=1}^N$.

**Classifier - Rejector Formulation** We first focus on building our L2D framework using the classifier-rejector approach (Madras et al., 2018a; Mozannar & Sontag, 2020a; Mozannar et al., 2023; Verma & Nalisnick, 2022a; Verma et al., 2023), first introduced by Cortes et al. (2016). This approach focuses on learning two models: a classifier denoted as $h : \mathcal{X} \to \mathcal{Y}$ and a rejector $r : \mathcal{X} \to \{0, 1, ..., J\}$. If $r(\boldsymbol{x}_i) = 0$, the classifier $h$ will make the decision on instance $i$; if $r(\boldsymbol{x}_i) = j$, the decision on instance $i$ will be deferred to expert $j$. We will consider the $0 - 1$ loss, as proposed by Verma et al. (2023), as the learning objective. According to this formulation, when the classifier makes the prediction (i.e., $r(\boldsymbol{x}) = 0$), the system incurs a loss of 1 if the classifier is incorrect. When expert $j$ makes the prediction (i.e., $r(\boldsymbol{x}) = j$), the system incurs a loss of 1 if the human is incorrect. Formally, the $0 - 1$ expected loss is defined as

$$L_{0-1}(h, r) = \mathbb{E}_{\boldsymbol{x}, y, \{m_j\}_{j=1}^J} \left[ \mathbb{I}[r(\boldsymbol{x}) = 0]\, \mathbb{I}[h(\boldsymbol{x}) \neq y] + \sum_{j=1}^J \mathbb{I}[r(\boldsymbol{x}) = j]\, \mathbb{I}[m_j(\boldsymbol{x}) \neq y] \right]. \tag{1}$$

Verma et al. (2023) demonstrate that the Bayes-optimal classifier and rejector, i.e., those that minimize $L_{0-1}$, satisfy

$$h^*(\boldsymbol{x}) = \arg\max_{y \in \mathcal{Y}} \mathbb{P}(y|\boldsymbol{x}) \tag{2}$$

$$r^*(\boldsymbol{x}) = \begin{cases} 0 & \text{if } \mathbb{P}(y = h^*(\boldsymbol{x})|\boldsymbol{x}) > \mathbb{P}(y = m_j(\boldsymbol{x})|\boldsymbol{x}),\ \forall j \in \{1, ..., J\} \\ \arg\max_{j \in \{1, ..., J\}} \mathbb{P}(y = m_j(\boldsymbol{x})|\boldsymbol{x}) & \text{otherwise,} \end{cases} \tag{3}$$

where $\mathbb{P}(y|\boldsymbol{x})$ is the probability of the label under the data generating process, and $\mathbb{P}(y = m_j|\boldsymbol{x})$ is the true probability that expert $j$ is be correct. As $L_{0-1}$ is non-convex, thus computationally hard to optimize, previous L2D work (Verma & Nalisnick, 2022a; Mozannar & Sontag, 2020a) focuses on the derivation of consistent convex surrogate losses, whose minimization results in $\hat{h}$ and $\hat{r}$ that approximate the Bayes optimal classifier-rejector pair. This approach assumes that $h$ and $r$ will be modelled by algorithms that fit the statistical query model (Kearns, 1998) (*e.g.*, neural networks, decision trees), whose learning process involves the empirical approximation of the expected value of the surrogate losses by using a training sample $S$. The minimization of the approximation of the expected value (*e.g.*, via gradient descent) produces a classifier $\hat{h}$ and a rejector $\hat{r}$ whose decisions are shown to converge to $h^*$ and $r^*$ as the amount of data increases.

In this work, rather than deriving a consistent surrogate loss for simultaneously training $h$ and $r$, we train both components separately. In the following paragraphs, we propose a training process for the classifier and the rejector under limited data availability, demonstrating their convergence to $h^*$ and $r^*$. For simplicity, we consider the binary classification case, where $\mathcal{Y} = \{0, 1\}$.

**Classifier Training**  To obtain $\hat{h}$, we can train a binary classifier using any proper binary composite loss with a well defined inverse link function $\psi^{-1}$, such as the logistic loss (Reid & Williamson, 2010), for which $\psi^{-1}(g) = 1/(1 + e^{-g})$. The empirical estimator of the expected value of the logistic loss is given by

$$\mathcal{L}_{\text{classifier}}\left(\{\boldsymbol{x}_i, y_i, c_i\}_{i=1}^N, g\right) = \frac{1}{N} \sum_{i=1}^N \left[ -y_i \log\left(\psi^{-1}\left(g(\boldsymbol{x}_i)\right)\right) - (1 - y_i) \log\left(1 - \psi^{-1}\left(g(\boldsymbol{x}_i)\right)\right) \right], \quad (4)$$

where $g(\boldsymbol{x}_i)$ is the score of sample $\boldsymbol{x}_i$. Proper binary composite losses ensure that $\mathbb{E}[\mathcal{L}]$ is minimized when $\psi^{-1}(g(\boldsymbol{x})) = \mathbb{P}(y = 1|\boldsymbol{x})$ (Buja et al., 2005; Reid & Williamson, 2010), meaning that minimization of this empirical loss will converge to $\hat{g}$ as $N \to \infty$, a scoring function producing calibrated estimates of $P(y = 1|\boldsymbol{x})$. As such, defining

$$\hat{h}(\boldsymbol{x}) = \begin{cases} 1 & \text{if } \psi^{-1}\left(\hat{g}(\boldsymbol{x})\right) > 0.5 \\ 0 & \text{otherwise}, \end{cases} \quad (5)$$

ensures that $\hat{h}$, in the large sample limit, will agree with the Bayes-optimal classifier $h^*$.

**Rejector Training**  In order to obtain $\hat{r}$, we resort to a human expertise model (HEM) with scoring function $g_\perp : \mathcal{X} \times \{1, ..., J\} \to \mathbb{R}$, predicting whether an expert is correct (1) or incorrect (0). By conditioning the model's output on the expert index, we can train this model using the training set $S$, which only contains one expert prediction per instance. Although it would be possible to train one scoring function per expert on the subset of the data with said expert's predictions (which would be similar to the approach of Verma et al. (2023)), we choose to use a single model for the entire team. In scenarios with limited data pertaining to each expert, it can be beneficial to model the team's behavior jointly, as there may not be enough data to train reliable models for every single expert. By conditioning the model's output on the expert's index, we still allow the model to adapt to each individual expert's decision-making processes, should that be beneficial. Note that, should additional information $\boldsymbol{e} \in \mathcal{E}$ be available to the experts (*e.g.*, ML model score), the HEM scoring function can take the form $g_\perp : \mathcal{X} \cup \{1, ..., J\} \cup \mathcal{E} \to \mathbb{R}$, thus conditioning the expert correctness probability estimate on all the information available to the expert. We obtain the optimal scoring function $\hat{g}_\perp$ by minimizing

$$\mathcal{L}_{\text{HEM}}\left(\{\boldsymbol{x}_i, y_i, m_{j,i}, j_i\}_{i=1}^N, g_\perp\right) = \frac{1}{N} \sum_{i=1}^N \left[ -\mathbb{I}[m_{j,i} = y] \log\left(\psi^{-1}\left(g_\perp(\boldsymbol{x}_i, j)\right)\right) \right.$$
$$\left. -\mathbb{I}[m_{j,i} \neq y] \log\left(1 - \psi^{-1}\left(g_\perp(\boldsymbol{x}_i, j)\right)\right) \right], \quad (6)$$

where $g_\perp(\boldsymbol{x}_i, j)$ is the score associated with deferring instance $\boldsymbol{x}_i$ to expert $j$. This allows us to obtain the estimates of $\mathbb{P}(y = m_j | \boldsymbol{x}, j)$, given by $\psi^{-1}\big(\hat{g}_\perp(\boldsymbol{x}, j)\big)$. As such, defining the rejector $\hat{r}$ as

$$\hat{r}(\boldsymbol{x}) = \begin{cases} 0 & \text{if } \max\{\psi^{-1}\big(\hat{g}(\boldsymbol{x})\big), 1 - \psi^{-1}\big(\hat{g}(\boldsymbol{x})\big)\} > \psi^{-1}\big(\hat{g}_\perp(\boldsymbol{x}, j)\big), \ \forall j \in \{1, ..., J\} \\ \underset{j \in \{1, ..., J\}}{\arg\max} \ \psi^{-1}\big(\hat{g}_\perp(\boldsymbol{x}, j)\big) & \text{otherwise,} \end{cases}$$

$$(7)$$

implies that $\hat{r}$ will agree with the Bayes-optimal rejector $r^*$, thus proving that our approach in modeling and training $\hat{r}$ and $\hat{h}$ under limited data availability yields a classifier-rejector pair that converges to $h^*$ and $r^*$, the minimizers of $L_{0-1}$,

## 3.2 Definition of Capacity Constraints

To shift focus to deferral under capacity constraints, we start by formalizing the work-capacity limitations of the expert team. Humans are limited in the number of instances they may process in any given time period (*e.g.*, work day). In real-world systems, human capacity must be applied over batches of instances, not over the whole dataset at once (*e.g.*, balancing the human workload over an entire month is not the same as balancing it daily). A real-world assignment system must then process instances taking into account the human limitations over a given "batch" of cases, corresponding to a pre-defined time period. We divide our dataset into several batches and, for each batch, define the maximum number of instances that can be processed by each expert. In any given dataset comprised of $N$ instances, divided into $n_{\text{batches}}$, capacity constraints can be represented by a vector $\boldsymbol{b}$, where component $b_i$ denotes which batch the instance $i \in \{1, ..., N\}$ belongs to, as well as a human capacity matrix $\boldsymbol{H} \in \mathbb{N}_0^{n_{\text{batches}} \times J}$, where element $H_{b,j}$ is a non-negative integer denoting the number of instances in batch $b$ that human expert $j$ can process.

## 3.3 Global Loss Minimization under Capacity Constraints

In the previous sections, we detailed our approach to train the classifier-rejector pair, showing that $\hat{h}$ and $\hat{r}$ converge to the Bayes-optimal $h^*$ and $r^*$. Note that this classifier-rejector formulation produces optimal point-wise solutions, disregarding any work-capacity limitations. Should an expert be more likely to be correct than all their colleagues and the ML classifier over the entire feature space, the set of optimal point-wise assignments would be to always defer instances to said expert, a clearly unfeasible solution in any real-world system.

A point-wise formulation of $\hat{h}$ and $\hat{r}$ is clearly inadequate to optimize the assignments over a set of instances while respecting expert capacity constraints. However, the scoring functions $\hat{g}$ and $\hat{g}_\perp$, obtained in the training steps detailed above, will still be of use in providing calibrated estimates of the probability of correctness. To formulate the assignment problem under capacity constraints, we consider that our objective is to maximize the expected probability of correctness over all instances to be deferred. Recall that $\mathcal{Y} = \{0, 1\}$ and consider the assignment decision for each instance $a_i \in \{1, ..., J + 2\}$, where $a_i = y + 1$, with $y \in \mathcal{Y}$, is an automatic prediction of class $y$ for instance $i$, whereas $a_i = j + 2$ denotes the decision to defer instance $i$ to the $j$th expert. The estimate of the probability of correctness for all possible assignments on a given instance is then given by

$$\hat{\mathbb{P}}(\text{correct} | \boldsymbol{x}_i, a_i) = \begin{cases} 1 - \psi^{-1}\big(\hat{g}(\boldsymbol{x}_i)\big) & \text{if } a_i = 1 \\ \psi^{-1}\big(\hat{g}(\boldsymbol{x}_i)\big) & \text{if } a_i = 2 \\ \psi^{-1}\big(\hat{g}_\perp(\boldsymbol{x}_i, j)\big) & \text{if } a_i = j + 2. \end{cases} \quad (8)$$

To represent the assignment decisions over a batch $b$ comprised of $n_B$ instances, without loss of generality indexed in $\{1, ..., n_B\}$, consider the $n_B \times (2 + J)$ matrix of assignments $\boldsymbol{A}$, where each element $A_{i,a_i}$ is a binary variable that denotes if the assignment decision $a_i$ is taken for instance $i$. The optimal set of assignments is

given by

$$
\boldsymbol{A}^* = \underset{A \in \{0,1\}^{n_B \times (2+J)}}{\arg\min} \sum_{i=1}^{n_B} \sum_{a_i=1}^{J+2} \hat{\mathbb{P}}(\text{correct}|\boldsymbol{x}_i, a_i) A_{i,a_i},
$$

$$
\text{s.t. } \sum_{i=1}^{n_B} A_{i,a_i} = H_{b,a_i}, \text{ for } a_i \in \{3, ..., J+2\},
$$

$$
\sum_{a_i=1}^{J+2} A_{i,a_i} = 1, \text{ for } i \in \{1, 2, ..., n_B\}.
$$

(9)

The first constraint refers to human capacity: with an equality constraint, the number of instances assigned to each decision-maker is predefined in the problem statement; this constraint may be changed to an inequality expressing the maximum number of assignments per decision-maker. The second constraint states that each instance must be assigned to one and only one decision-maker. We solve the assignment problem 9 using the constraint programming solver CP-SAT from Google Research's OR-Tools (Perron & Didier, 2023). Finding solutions to these combinatorial problems is not trivial, as solution times grow exponentially with the problem's size (Huberman et al., 1997). CP-SAT is a parallel portfolio solver, leveraging the available processor cores to run different solution search strategies on each. By doing so, portfolio solvers aim to more efficiently cover the solution-space, under the assumption that different algorithms, using different heuristics, may compliment each other's weaknesses (Huberman et al., 1997; Gomes & Selman, 2001). This approach proves effective as selecting the optimal search algorithm for a given constraint problem is considered a non-trivial task. We selected CP-SAT in our implementation as it has been repeatedly shown to be the best-performing publicly available solver in a wide array of constraint programming problems in the MiniZinc challenge (Stuckey et al., 2014), where solvers are tasked to finding the optimal or a near-optimal solution within a time-limit. While this solver can find feasible solutions relatively quickly, proving a solution's optimality can take an extremely long time, meaning that in practice it is often better to set a time limit, having the system return the best solution it found so far. On our machine (Intel(R) Xeon(R) Gold 5120 CPU @ 2.20GHz), no significant improvements were obtained for timeout limits above 60 seconds.

### 3.4 Cost-Sensitive Learning

Finally, having discussed how to optimize for the 0-1 loss under capacity constraints, we now focus on adapting our method to work under an arbitrary cost structure. To do so, we follow an instance re-weighting approach (Zadrozny et al., 2003; Elkan, 2001), where each point-wise loss over the training set is multiplied by the cost $c_i$ associated with said instance. This guarantees that the minimization of the surrogate losses used to train $g$ and $g_\perp$ will result in scoring functions that minimize the misclassification cost instead of the error-rate. A more detailed description of the re-weighting approach follows.

Training a classifier $h$ with score function $g$ involves approximating the expected value of its surrogate loss function $\ell$ by the empirical average of the point-wise losses over the training set,

$$
\mathbb{E}_{(\boldsymbol{x},y)\sim D}[\ell(\boldsymbol{x}, g(\boldsymbol{x}), y)] \approx \frac{1}{N} \sum_{i}^{N} \ell(\boldsymbol{x}_i, g(\boldsymbol{x}_i), y_i),
$$

(10)

where $D$ denotes the data distribution. Note, however, that instances may have different misclassification costs, in which case a surrogate for the 0-1 loss is not adequate. Assuming that each instance $\boldsymbol{x}_i$ has an associated misclassification cost $c_i$, the goal is then to learn a classifier that minimizes the expected cost, $\mathbb{E}_{\boldsymbol{x},y,c\sim D}[c\,\mathbb{I}(h(\boldsymbol{x}) \neq y)]$. Minimizing surrogates to the 0-1 loss ensures that we minimize $\mathbb{E}_{\boldsymbol{x},y,c\sim D}[\mathbb{I}(h(\boldsymbol{x}) \neq y)]$, which is misaligned with our objective. Zadrozny et al. (2003) show that if we have examples drawn from a different distribution,

$$
\tilde{D}(\boldsymbol{x}, y, c) = \frac{c}{\mathbb{E}_{c\sim D}[c]} D(\boldsymbol{x}, y, c),
$$

(11)

then

$$\mathbb{E}_{\boldsymbol{x},y,c\sim\tilde{D}}[\mathbb{I}(h(\boldsymbol{x})\neq y)] = \frac{1}{\mathbb{E}_{c\sim D}[c]}\mathbb{E}_{\boldsymbol{x},y,c\sim D}[c\,\mathbb{I}(h(\boldsymbol{x})\neq y)]. \tag{12}$$

Equation 12 shows that selecting the decision rule $h$ to minimize the error rate under $\tilde{D}$ is equivalent to selecting $h$ to minimize the expected misclassification cost under $D$. This means we can obtain our desired classifier by training it under the distribution $\tilde{D}$, using the log-loss. To train a classifier under $\tilde{D}$, a common approach (Zadrozny et al., 2003; Elkan, 2001) is to re-weight the instances according to their costs. In this way, we obtain a classifier $\hat{h}$ that prioritizes correctness according to the instances' weights, by minimizing the empirical estimate of the misclassification cost:

$$\frac{1}{N\,\mathbb{E}_{c\sim D}[c]}\sum_{i=1}^{N}c_i\ell(\boldsymbol{x}_i, g(\boldsymbol{x}_i), y_i). \tag{13}$$

As such, we just have to reweight the instances in $\mathcal{L}_{\text{classifier}}$, where $\ell$ is the point-wise log-loss (see Eq. 4), when training the scoring function $g$. We follow the same approach with respect to the rejector, reweighting the instances in $\mathcal{L}_{\text{HEM}}$ (see Eq. 6) when training $g_\perp$. The re-weighted empirical losses are given by

$$\mathcal{L}'_{\text{classifier}}\left(\{\boldsymbol{x}_i, y_i, c_i\}_{i=1}^{N}, g\right) = \frac{1}{N}\sum_{i=1}^{N}c_i\left[-y_i\log\left(\psi^{-1}\big(g(\boldsymbol{x}_i)\big)\right) - (1-y_i)\log\left(1-\psi^{-1}\big(g(\boldsymbol{x}_i)\big)\right)\right], \tag{14}$$

$$\begin{aligned}\mathcal{L}'_{\text{HEM}}\left(\{\boldsymbol{x}_i, y_i, c_i, m_{j,i}, j_i\}_{i=1}^{N}, g_\perp\right) = \frac{1}{N}\sum_{i=1}^{N}c_i\Big[&-\mathbb{I}[m_{j,i}=y]\log\left(\psi^{-1}\big(g_\perp(\boldsymbol{x}_i, j)\big)\right)\\ &-\mathbb{I}[m_{j,i}\neq y]\log\left(1-\psi^{-1}\big(g_\perp(\boldsymbol{x}_i, j)\big)\right)\Big].\end{aligned} \tag{15}$$

The full algorithm is described in the following pseudo-code blocks. Note that, for the training process, the function "Minimize-Loss" represents any optimization algorithm that minimizes an empirical loss (*e.g.*, gradient descent, gradient boosting).

---

**Algorithm 1** Pseudo-Code for DeCCaF - Training

---

1: **Input:** Training data $S = \{\boldsymbol{x}_i, y_i, m_{j,i}, c_i\}_{i=1}^{N}$
2: Initialize $g$, $g_\perp$                               ▷ Initial scoring functions (*e.g.*, NN with random weights)
3: $\hat{g} \leftarrow$ Minimize-Loss$\left(\mathcal{L}'_{\text{classifier}}\left(\{\boldsymbol{x}_i, y_i, c_i\}_{i=1}^{N}, g\right)\right)$                   ▷ see Eq. 14
4: $\hat{g}_\perp \leftarrow$ Minimize-Loss$\left(\mathcal{L}'_{\text{HEM}}\left(\{\boldsymbol{x}_i, y_i, c_i, m_{j,i}\}_{i=1}^{N}, g_\perp\right)\right)$           ▷ see Eq. 15

---

## 4 Experiments

### 4.1 Experimental Setup

**Dataset** As the base dataset, we choose to use the publicly available bank-account-fraud dataset (Jesus et al., 2022) (Version 1). This tabular dataset is comprised of one million synthetically generated bank account opening applications, where the label denotes whether the instance is fraudulent (1) or legitimate (0). The features of each instance contain information about the application and the applicant, and the task of a decision maker (automated or human) is to either accept (0) or reject (1) it.

These applications were generated based on anonymized real-world bank account applications, and, as such, this dataset poses challenges typical of real-world high-stakes applications. Firstly, there is a high class imbalance, with 1.1% fraud prevalence over the entire dataset. Furthermore, there are changes in the data distribution over time, often referred to as concept drift (Gama et al., 2014), which can severely impact the predictive performance of ML Models.

---

**Algorithm 2** Pseudo-Code for DeCCaF - Inference for Batch $b$

---

1: **Input:** Instances $\{\boldsymbol{x}_i\}_{i=1}^{n_B}$, Batch Capacity Constraints $\{\boldsymbol{H}_{b,j}\}_{j=i}^{J}$    $\triangleright$ $\boldsymbol{H} \in \mathbb{N}_0^{n_{\text{batches}} \times J}$, see Section 3.2
2: Initialize $\boldsymbol{P} \in \mathbb{R}^{n_B \times (J+2)}$                  $\triangleright$ Stores $\hat{\mathbb{P}}(\text{correct}|\boldsymbol{x}_i, a_i)$, see Eq. 8
3: Initialize $\boldsymbol{A} \in \{0,1\}^{n_B \times (J+2)}$    $\triangleright$ Stores $A_{i,a_i}$ denoting if decision $a_i$ is taken for instance i, see Eq. 9
4: **for** $i \leftarrow \{1,..,n_B\}$ **do**
5:      **for** $a_i \leftarrow \{1,..,J+2\}$ **do**
6:          **if** $a_i = 1$ **then**
7:              $P_{i,a_i} \leftarrow 1 - \psi^{-1}\big(\hat{g}(\boldsymbol{x}_i)\big)$
8:          **else if** $a_i = 2$ **then**
9:              $P_{i,a_i} \leftarrow \psi^{-1}\big(\hat{g}(\boldsymbol{x}_i)\big)$
10:          **else**
11:              $P_{i,a_i} \leftarrow \psi^{-1}\big(\hat{g}_\perp(\boldsymbol{x}_i, a_i - 2)\big)$
12:          **end if**
13:      **end for**
14: **end for**
15: $\boldsymbol{A}^* \leftarrow \text{CPSolver}\big(\boldsymbol{P}, \{\boldsymbol{H}_{b,j}\}_{j=i}^{J}\big)$                $\triangleright$ Optimization Problem in (7)

---

**Alert Review Setup** There are several possible ways for models and humans to cooperate. In previous L2D research, it is a common assumption that any instance can be deferred to either the model or the expert team. However, in real-world settings, due to limitations in human work-capacity, it is common to use an Alert Model to screen instances, raising alerts that are then reviewed by human experts (De-Arteaga et al., 2020; Han et al., 2020). In an alert review setting, humans only predict on a fraction of the feature space, that is, the instances flagged by the Alert Model. We will train a L2D system to work in tandem with the Alert Model, by deferring the alerts in an intelligent manner. We calculate the Alert Model's alert rate $a_r$, that is, the fraction of instances flagged for human review, by determining the FPR of the Alert Model in validation. We create distinct alert review scenarios by varying the alert rate $a_r \in \{5\%\text{FPR}, 15\%\text{FPR}\}$.

**Alert Model Training** We train the Alert Model to predict the fraud label on the first three months of the dataset, validating its performance on the fourth month. We use the LightGBM (Ke et al., 2017) algorithm, due to its proven high performance on tabular data (Shwartz-Ziv & Armon, 2022; Borisov et al., 2022). Details on the training process of the classifier are given in Section A.1 of the Appendix.

**Optimization Objective** This is a cost-sensitive task, where the cost of a false positive (incorrectly rejecting a legitimate application) must be weighed against the cost of a false negative (incorrectly accepting a fraudulent application). Due to the low fraud prevalence, metrics such as accuracy are not adequate to measure the performance of both ML models and deferral systems. The optimization objective used by Jesus et al. (2022) is a Neyman-Pearson criterion, in this case, maximizing recall at 5% false positive rate (FPR), which establishes an implicit relationship between the costs of false positive and a false negative errors. However, for the cost-sensitive learning method described in Section 3.4, we need to have access to the explicit cost structure of the task at hand.

In a cost-sensitive task, the optimization goal is to obtain a set of predictions $\hat{y}$ that minimize the expected misclassification cost $\mathbb{E}[\mathcal{C}]$. Assuming that correct classifications carry no cost, the relevant parameter is the ratio $\lambda = c_{\text{FP}}/c_{\text{FN}}$, were $c_{\text{FP}}$ and $c_{\text{FN}}$ are the costs of false positive and false negative errors, respectively. The objective is thus

$$\frac{1}{N} \sum_{i=1}^{N} \big[\lambda \mathbb{I}[y_i = 0 \wedge \hat{y}_i = 1] + \mathbb{I}[y_i = 1 \wedge \hat{y}_i = 0]\big]. \tag{16}$$

Minimizing this quantity is equivalent to minimizing the average cost, as division by a constant will not affect the ranking of different assignments. As such, all that remains to be established is a relationship between the Neyman-Pearson criterion and the value of $\lambda$. To to do so, we follow the approach detailed in Section A.2 of the Appendix, which yields the theoretical value $\lambda_t = 0.057$. To test performance under variable cost structures, we will conduct experiments for the values $\lambda \in \{\lambda_t/5, \lambda_t, 5\lambda_t\}$. These alternative scenarios are not

strictly comparable, as if the cost structure were $\lambda \in \{\lambda_t/5, 5\lambda_t\}$, the optimization of the Alert Model would not correspond to maximizing the recall at 5% FPR. Nevertheless, we choose to introduce these variations to test the impact on system performance of changing the cost ratio $\lambda$. Combining the different alert rates with these values of $\lambda$, we obtain 6 distinct text scenarios.

**Synthetic Expert Decision Generation**  Our expert generation approach is based on *instance-dependent noise*, in order to obtain more realistic experts, whose probability of error varies with the properties of each instance. We generate synthetic predictions by flipping each label $y_i$ with probability $\mathbb{P}(m_{j,i} \neq y_i | \boldsymbol{x}_i, y_i)$. In some HAIC systems, the model score for a given instance may also be shown to the expert (Amarasinghe et al., 2022; De-Arteaga et al., 2020; Levy et al., 2021), so an expert's decision may also depend on an ML model score $m(\boldsymbol{x}_i)$. We define the expert's probabilities of error, for a given instance, as a function of a pre-processed version of its features $\bar{\boldsymbol{x}}_i$ and the Alert Model's score $M(\boldsymbol{x}_i)$, so that the feature scale does not impact the relative importance of each quantity. The probabilities of error are given by

$$
\begin{aligned}
\mathbb{P}(m_{j,i} = 1 | y_i = 0, \boldsymbol{x}_i) &= \sigma\left(\beta_0 - \alpha \frac{\boldsymbol{w} \cdot \bar{\boldsymbol{x}}_i + w_M M(\boldsymbol{x}_i)}{\sqrt{\boldsymbol{w} \cdot \boldsymbol{w} + w_M^2}}\right) \\
\mathbb{P}(m_{j,i} = 0 | y_i = 1, \boldsymbol{x}_i) &= \sigma\left(\beta_1 + \alpha \frac{\boldsymbol{w} \cdot \bar{\boldsymbol{x}}_i + \boldsymbol{w}_M M(\boldsymbol{x}_i)}{\sqrt{\boldsymbol{w} \cdot \boldsymbol{w} + w_M^2}}\right),
\end{aligned}
\tag{17}
$$

where $\sigma$ denotes a sigmoid function. Each expert's probabilities of the two types of error depend on five parameters: $\beta_0, \beta_1, \alpha, \boldsymbol{w}$, and $w_M$. The weight vector $\boldsymbol{w}$ embodies a relation between the features and the probability of error. The feature weights are normalized so that we can separately control, via $\alpha$, the overall magnitude of the dependence of the probability of error on the instance's features. The values of $\beta_1$ and $\beta_0$ control the base probability of error. The motivation for this approach is explained further in section C.1 of the Appendix.

The expected cost resulting from an expert's decisions is given by

$$
\begin{aligned}
\mathbb{E}[\mathcal{C}]_j &= \mathbb{E}_y\left[\lambda \mathbb{P}(m_j = 1 \wedge y = 0) + \mathbb{P}(m_j = 0 \wedge y = 1)\right] \\
&\approx \frac{1}{N}\sum_{i=1}^{N}\left[\lambda \mathbb{P}(m_{j,i} = 1 | y_i = 0)\mathbb{P}(y_i = 0) + \mathbb{P}(m_{j,i} = 0 | y_i = 1)\mathbb{P}(y_i = 1)\right].
\end{aligned}
\tag{18}
$$

For our setup to be realistic, we assume an expert's decisions must, on average, incur a lower cost than simply automatically rejecting all flagged transactions. Otherwise, assuming random assignment, having that expert in the human team would harm the performance of the system as a whole. As the expert's average misclassification cost is dependent on the prevalence $\mathbb{P}(y = 1)$ and cost structure as defined by $\lambda$, a team of experts with the exact same parameters will have different expected misclassification costs, depending on the alert review scenario in question. For this reason, in each of the six aforementioned settings, we generate a different team of 9 synthetic experts by first fixing their feature weights $\boldsymbol{w}$, and sampling their expected misclassification cost. Then, we calculate the values of $\beta_0, \beta_1$ that achieve the sampled misclassification cost, thus obtaining a team of complex synthetic experts with desirable properties within each scenario. Further details on the sampling and expert generation process, as well as a description of each of the expert teams' properties are available in Sections C.2 and C.3 of the Appendix.

**Expert Data Availability**  To generate realistic expert data availability, we assume that the Alert Model is deployed in months 4-7, and that each alert is deferred to a randomly chosen expert. This generates a history of expert decisions with only one expert's prediction per instance. To introduce variability to the L2D training process, the random distribution of cases throughout experts was done with 5 different random seeds per scenario. For each of the three 5% FPR alert rate settings, this results in a set of 2.9K predictions per expert. So that all settings have the same amount of data, for the 15% FPR alert rate scenarios, we also sample 2.9K predictions from each of the 10 experts.

**Expert Capacity Constraints**  In our experiments, we want to reliably measure the ability of our method to optimize the distribution of instances throughout the human team and the classifier $h$. In real-world

Table 1: Distribution of Expert Decision Outcomes

| SCENARIO | | DECISION OUTCOME(%) | | | |
|---|---|---|---|---|---|
| $a_r$ | $\lambda$ | fp | fn | tp | tn |
| 0.05 | 0.0114 | 30.2 | 0.3 | 11.1 | 58.4 |
| 0.05 | 0.057 | 25.0 | 1.9 | 9.4 | 63.7 |
| 0.05 | 0.285 | 16.9 | 5.6 | 5.9 | 71.6 |
| 0.15 | 0.0114 | 34.4 | 0.3 | 5.5 | 59.9 |
| 0.15 | 0.057 | 30.6 | 1.4 | 4.2 | 63.8 |
| 0.15 | 0.285 | 10.4 | 2.7 | 3.1 | 83.8 |

scenarios, the cost of querying $h$ is much lower than that of querying a human expert, as the work-capacity of the classifier is virtually limitless. However, in order to ensure that the assignment systems are able to reliably model the human behavior of all experts, we will force the assigner to distribute the instances throughout the expert team and the classifier $h$ in equal amounts. As the expert teams are comprised of 9 experts, this means that 1/10 of the test set will be deferred to each expert/classifier. To introduce variability, we also create four distinct test settings where each expert/classifier has a different capacity, by sampling the values from $\mathcal{N}(N_{test}/10, N_{test}/50)$. As such, for each scenario defined by the $(a_r, \lambda)$ pair, there are a total of 25 test variations, which result from combining each of the 5 training seeds with the 5 different capacity constraint settings.

## 4.2 Baselines

**One vs. All (OvA)** For an L2D baseline, we use the multi-expert learning to defer OvA algorithm, proposed by Verma et al. (2023). This method originally takes training samples of the form $D_i = \{\boldsymbol{x}_i, y_i, m_{i,1}, ..., m_{j,i}\}$ and assumes the existence of a set of every expert's predictions for each training instance; however, this is not strictly necessary.

The OvA model relies on creating a classifier $h : \mathcal{X} \to \mathcal{Y}$ and a rejector $r : \mathcal{X} \to \{0, 1, ..., J\}$. If $r(\boldsymbol{x}_i) = 0$, the classifier makes the decision; if $r(\boldsymbol{x}_i) = j$, the decision is deferred to the $j$th expert. The classifier is composed of K functions: $g_k : \mathcal{X} \to \mathbb{R}$, for $k \in \{1, ..., K\}$, where $k$ denotes the class index. These are related to the probability that an instance belongs to class $k$. The rejector, similarly, is composed of J functions: $g_{\perp,j} : \mathcal{X} \to \mathbb{R}$ for $j \in \{1, ..., J\}$, which are related to the probability that expert $j$ will make the correct decision regarding said instance. The authors propose combining the functions $g_1, ..., g_K, g_{\perp,1}, ..., g_{\perp,J}$ in an OvA surrogate for the 0-1 loss. The OvA multi-expert L2D surrogate is defined as:

$$\Psi_{\text{OvA}} = \Phi[g_y(\boldsymbol{x})] + \sum_{y' \in \mathcal{Y}, y' \neq y} \Phi[g'_y(\boldsymbol{x})] + \sum_{j=1}^{J} \Phi[-g_{\perp,j}(\boldsymbol{x})] + \sum_{j=1}^{J} \mathbb{I}[m_j = y](\Phi[g_{\perp,j}(\boldsymbol{x})] - \Phi[-g_{\perp,j}(\boldsymbol{x})]),$$

where $\Phi : \{\pm 1\} \times \mathbb{R} \to \mathbb{R}_+$ is a strictly proper binary surrogate loss. (The authors also propose using the logistic loss.) Verma et al. (2023) then prove that the minimizer of the pointwise inner risk of this surrogate loss can be analyzed in terms of the pointwise minimizer of the risk for each of the $K + J$ underlying OvA binary classification problems, concluding that the minimizer of the pointwise inner $\Psi_{\text{OvA}}$-risk, $\boldsymbol{g}^*$, is comprised of the minimizer of the inner $\Phi$-risk for each $i$th binary classification problem, $g_i^*$. As such, in a scenario where only one expert's prediction is associated with each instance, each binary classifier $g_{\perp,j}$ can be trained independently of the others. By training each binary classifier $g_{\perp,j}$ with the subset of the training sample containing expert $j$'s predictions, we obtain the best possible estimates of each pointwise inner $\Phi$-risk minimizer $g_i^*$ given the available data. To adapt the OvA method to a cost-sensitive scenario, we can again use the rescaling approach detailed in Section 3.4, minimizing the expected misclassification cost.

The OvA method does not support assignment with capacity constraints, as such, we proceed similarly to Verma et al. (2023), by considering the maximum out of the rejection classifiers' predictions. If the capacity of the selected expert is exhausted, deferral is done to the second highest scoring expert, and so on.

**Random Assignment** In this baseline, which aims to represent the average performance of the expert/model team under the test conditions, alert deferral is done randomly throughout the experts/model until each of their capacity constraints are met.

**Only Classifier (OC)** In this baseline, all final decisions on the alerts are made by the classifier $h$.

**Full Rejection (FR)** All alerts are automatically rejected.

### 4.3 Deferral System Training

For the classifier $h$ and the HEM, we again use a LightGBM model, trained on the sample of alerts raised over months 4 to 6, and validated on month 7. The models were selected in order to minimize the weighted log-loss, where the weight for a label-positive instance is $c_i = 1$, and the weight for a label-negative instance is given by $c_i = \lambda$.

For the OvA method, we follow the process detailed in Section 4.2, first splitting the training set into the data pertaining to each expert's prediction to obtain each scoring function $g_{\perp,j}$. In the binary case, only one classifier scoring function $g$ is needed for the OvA approach. As each scoring function is trained independently, the optimal classifier $h$ is obtained in the same manner as our method. As such, we will use the same classifier $h$ for both deferral systems. Details on the training process and hyper-parameter selection are available in Section D of the Appendix.

## 5 Results

### 5.1 Classifier h - Performance and Calibration

We first evaluate the predictive performance and calibration of the classifier $h$. We assess how well the classifier is able to rank probabilities of correctness, and then evaluate its calibration, by using the *expected calibration error* (Guo et al., 2017). Note that each classifier had a distinct training process, with varying sample weights $c_i$ dependent on $\lambda$. As such, these measures must be calculated under the re-weighted data distribution $\tilde{D}$ (see section 3.4). In Table 2, we show that the classifier $h$ is able to rank instances according to their probability of belonging to the positive class.

Table 2: Expected calibration error (ECE) and ROC-AUC for classifier $h$ for all testing scenarios, denoted by the {alert rate $(a_r)$, cost-structure $(\lambda)$} pairs – classifier $h$ is able to reliably estimate $\mathbb{P}(y_i = 1)$

| SCENARIO | | CLASSIFIER $h$ | |
| --- | --- | --- | --- |
| $a_r$ | $\lambda$ | ROC-AUC | ECE (%) |
| 0.05 | 0.011 | 0.70 | 1.1 |
| 0.05 | 0.057 | 0.71 | 4.8 |
| 0.05 | 0.285 | 0.71 | 4.6 |
| 0.15 | 0.011 | 0.76 | 3.8 |
| 0.15 | 0.057 | 0.75 | 4.2 |
| 0.15 | 0.285 | 0.73 | 3.3 |

### 5.2 Expert Decision Modeling - Performance and Calibration

We now evaluate the probability ranking and calibration of the models that estimate expert correctness. We will again check that the scoring functions $g_\perp$ (HEM) and $g_{\perp,j}$ (OvA) are able to model individual expert correctness by using the ROC-AUC, then checking for calibration. For each training random seed, the ROC-AUC was calculated for every individual expert's decisions. Again, these measures were calculated under the data distribution $\tilde{D}$. In the top row of Figure 2, we observe that the distribution of ROC-AUC is similar across both methods, with consistent overlap of error bars, despite a superior average across almost

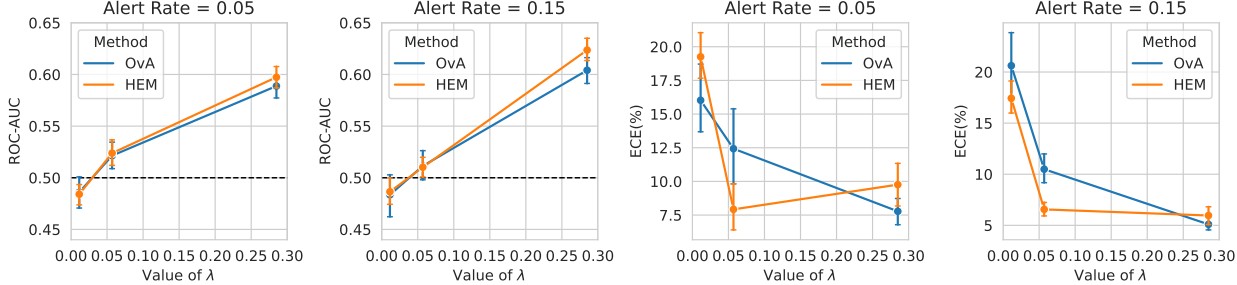

Figure 2: Mean ECE and ROC-AUC for estimates of $\mathbb{P}(y_i = m_{j,i})$. Values are calculated for each expert $j$ and averaged, with error bars representing a 95(%) confidence interval. Both methods obtain similar ROC-AUC, however, the value of $\lambda$ has significant impact on their ranking in terms of calibration, showing the importance of testing L2D methods under a wide variety of cost-structures.

all scenarios for the HEM method. It is notable, however, that for both alert rates, the value of the ROC-AUC consistently falls below 0.50 when $\lambda = 0.0114$, indicating that these models are not able to reliably rank probabilities of correctness of the expert team under these conditions. In these scenarios, however, L2D methods can still theoretically achieve improvements in performance by choosing which instances should be deferred to the classifier $h$, whose probabilities of correctness have been shown to be reliably modeled. In the bottom row of Figure 2, we observe that HEM achieves significantly lower ECE for $\lambda = 0.057$, with the reweighted OvA baseline outperforming HEM in scenarios with $\lambda = 0.285$. This demonstrates how the performance ranking of different L2D methods can change significantly by altering the data distribution and cost-structure of the experimental setting.

## 5.3 Deferral under Capacity Constraints

In this section, we evaluate the quality of deferral as measured by the average misclassification cost per 100 instances. The average experimental results are displayed in Table 3 with 95% confidence intervals, where we refer to our method as DeCCaF. We observe that, on average, both L2D methods outperform the non-L2D approaches, and that DeCCaF is significantly better in most scenarios. As the number of test seeds (25) is relatively low, the approximation of the 95% confidence interval may be unreliable, as the value shown is derived based on the central limit theorem.

Table 3: Expected misclassification cost per 100 instances ($\mathbb{E}[\mathcal{C}]/100$, assuming $c_{\text{FP}} = \lambda, c_{\text{FN}} = 1$) for each $\{a_r, \lambda\}$ pair. In each row, values are averaged across all 25 test variations, and displayed with 95% confidence intervals. FR and OC represent the "Full Rejection" and "Only Classifier" baselines.

| SCENARIO | | DEFERRAL STRATEGY | | | | |
|---|---|---|---|---|---|---|
| $a_r$ | $\lambda$ | FR | OC | Random | OvA | DeCCaF |
| 0.05 | 0.0114 | 0.96 | 0.96 | $0.80_{\pm 0.08}$ | $0.84_{\pm 0.06}$ | $\mathbf{0.79}_{\pm 0.04}$ |
| 0.05 | 0.057 | 4.79 | 4.58 | $4.0_{\pm 0.2}$ | $3.63_{\pm 0.07}$ | $\mathbf{3.4}_{\pm 0.04}$ |
| 0.05 | 0.285 | 23.95 | 14.08 | $12.2_{\pm 0.2}$ | $11.13_{\pm 0.04}$ | $\mathbf{10.2}_{\pm 0.2}$ |
| 0.15 | 0.0114 | 1.05 | 1.07 | $0.87_{\pm 0.05}$ | $0.87_{\pm 0.02}$ | $\mathbf{0.74}_{\pm 0.03}$ |
| 0.15 | 0.057 | 5.27 | 4.01 | $3.5_{\pm 0.1}$ | $3.55_{\pm 0.03}$ | $\mathbf{3.19}_{\pm 0.03}$ |
| 0.15 | 0.285 | 26.36 | 6.93 | $6.0_{\pm 0.1}$ | $5.23_{\pm 0.08}$ | $\mathbf{4.85}_{\pm 0.09}$ |

In Table 4, we report the percentage of test variations, for each scenario, in which our method outperforms other deferral baselines. These results demonstrate that, across all scenarios, DeCCaF outperforms the full rejection and the "Only Classifier" baseline. Notably, the relative performance of DeCCaF when compared to random assignment changes drastically depending on the scenario properties, performing no better than

Table 4: Percentage of the 25 test variations, for each $\{a_r, \lambda\}$ pair, where DeCCaF beats other methods. According to a binomial statistical significance test, with $\alpha = 0.05$, DeCCaF is significantly better than other methods for values $\in [0.68, 1]$, while values $\in [0.28, 0.68]$ mean that we cannot conclude which method is best. Across all 6 scenarios, DeCCaF is shown to be superior in 5, while the comparison is inconclusive in one.

| SCENARIO | | DEFERRAL STRATEGY | | | |
|---|---|---|---|---|---|
| $a_r$ | $\lambda$ | OvA | Random | FR | OC |
| 0.05 | 0.0114 | 0.52 | 0.56 | 0.88 | 0.88 |
| 0.05 | 0.057 | 0.76 | 1.00 | 1.00 | 1.00 |
| 0.05 | 0.285 | 0.96 | 1.00 | 1.00 | 1.00 |
| 0.15 | 0.0114 | 0.84 | 0.88 | 1.00 | 1.00 |
| 0.15 | 0.057 | 1.00 | 0.96 | 1.00 | 1.00 |
| 0.15 | 0.285 | 0.96 | 1.00 | 1.00 | 1.00 |

random assignment for $a_r = 5\%FPR, \lambda = 0.0114$. This again demonstrates how the cost structure and distribution of expert performances has significant impact on the performance of L2D systems. Nevertheless, we observe that DeCCaF performs significantly better than the OvA baseline in 5 out of 6 scenarios, outperforming in both scenarios with $\lambda = \lambda_t$. Note that the only scenario where the comparison between DeCCaF and other baselines is inconclusive corresponds to the most imbalanced scenario, where the prevalence is higher, and the cost of false positives is lowest. We therefore conclude that there are benefits in jointly modeling the human decision-making processes.

**Varying Data Avaliability** Finally, we assess the impact of the amount of training data by repeating the experiments for $\lambda = \lambda_t = 0.057$, but training the OvA and DeCCaF methods with less data.

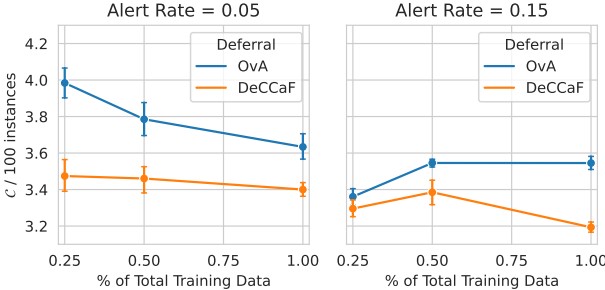

Figure 3: Expected Misclassification Cost per 100 instances ($\mathbb{E}[\mathcal{C}]/100$, assuming $c_{\text{FP}} = \lambda, c_{\text{FN}} = 1$) for $a_r \in \{0.05, 0.15\}$, $\lambda = 0.057$, and different amounts of training data. In each point, values are averaged across all 25 test variations, and displayed with 95% confidence intervals - DeCCaF remains significantly better in most scenarios.

In Figure 3, we show the variations in misclassification cost as a result of the undersampling of training data. As expected, both methods are significantly impacted by reducing the amount of training data. We again observe that DeCCaF performs either similarly to, or outperforms the OvA baseline.

## 6 Conclusions and Future Work

In this work, we expand multi-expert L2D research to consider several real-world issues that limit adoption of these systems. We consider limited data availability, the existence of human expert work-capacity constraints, cost-sensitive optimization objectives, and significant class imbalance, which are key challenges posed by many high-stakes classification tasks.

We propose a novel architecture that aims to better model human behaviour and to globally optimize the system's performance under capacity constraints. We conducted constrained deferral under a wide variety of training and testing conditions, in a realistic cost-sensitive classification task, empirically demonstrating that variations in the cost structure, data distribution, and human behavior can have significant impact on the relative performance of deferral strategies. We demonstrate that DeCCaF performs significantly better than the baselines in a wide array of testing scenarios, showing that there are benefits to jointly modeling the expert team.

For future work, we plan on testing the addition of fairness incentives to our misclassification cost optimization method, to study the impact this system may have in ensuring fairness. For scenarios where the misclassification costs are instance-specific (*i.e.*, transaction fraud), we will also study the possibility of direct cost estimation by using regression models instead of classification.

Finally, it is important to consider the ethical implications of adopting these systems in real-world scenarios, as these may impact the livelihood of the human experts involved in the deferral process. In a system without intelligent assignment, cases are distributed randomly throughout the human team, ensuring an i.i.d. data distribution for each expert. If a multi-expert L2D system were to be adopted, the subset of cases deferred to each expert would follow a separate distribution. If we consider a highly skilled expert with very high performance throughout the feature space, this system could choose to assign the hardest cases to said expert, damaging their performance. As the performance of analysts is routinely evaluated, this could create unfair disparities across the human expert team, with certain analyst's performance being degraded while others could be inflated. To tackle this issue, we could periodically assign randomly selected instances to each expert, in order to evaluate them in an i.i.d set, which would also be useful data to retrain the system, as human behaviour may change over time. We hope this work promotes further research into L2D methods in realistic settings that consider the key limitations that inhibit the adoption of previous methods in real-world applications.

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

## Notation

$h$ Classifier

$\hat{h}$ Optimized Classifier

$h^*$ Bayes Optimal Classifier

$\hat{g}$ Optimized classifier scoring function

$r$ Classifier

$\hat{r}$ Optimized rejector

$r^*$ Bayes Optimal Rejector

$\hat{g}_\perp$ Optimized rejector scoring function

**b** batch vector, where component $b_i$ denotes which batch instance $i$ belongs to

**H** Capacity matrix, where component $H_{b,j}$ denotes the number of instances in batch $b$ the human expert $j$ can process

$a_i$ Assignment decision for instance $i$. $a_i \in \{1, ..., J+2\}$, where $a_i = y + 1$, with $y \in \mathcal{Y}$, is an automatic prediction of class $y$ for instance $i$, whereas $a_i = j + 2$ denotes the decision to defer instance $i$ to the $j$th expert.

**A** matrix of assignments , where each element $A_{i,a_i}$ is a binary variable that denotes if the assignment decision $a_i$ is taken for instance $i$.

**A**$^*$ matrix of optimal assignments , where each element $A^*_{i,a_i}$ is a binary variable that denotes if the assignment decision $a_i$ is taken for instance $i$.

## A Experimental Setting

### A.1 Alert Model

As detailed in Section 4.3, our Alert Model is a LightGBM (Ke et al., 2017) classifier. The model was trained on the first 3 months of the BAF dataset, and validated on the fourth month. The model is trained by minimizing binary cross-entropy loss. The choice of hyperparameters is defined through Bayesian search (Akiba et al., 2019) on an extensive grid, for 100 trials, with validation done on the 4th month, where the optimization objective is to maximize recall at 5% false positive rate in validation. In Table 5 we present the hyperparameter search space used, as well as the parameters of the selected model.

Table 5: Alert Model: LightGBM hyperparameter search space

| HYPERPARAMETER | VALUES OR INTERVAL | DIST. | SELECTED |
|---|---|---|---|
| boosting_type | "goss" | | "goss" |
| enable_bundle | False | | False |
| n_estimators | [50,5000] | Log | 94 |
| max_depth | [2,20] | Unif. | 2 |
| num_leaves | [10,1000] | Log | 145 |
| min_child_samples | [5,500] | Log | 59 |
| learning_rate | [0.01, 0.5] | Log | 0.3031421 |
| reg_alpha | [0.0001, 0.1] | Log | 0.0012637 |
| reg_lambda | [0.0001, 0.1] | Log | 0.0017007 |

This model yielded a recall of 57.9% in validation, for a threshold $t = 0.050969$, defined to obtain a 5% FPR in validation. In the deployment split (months 4 to 8), used to train and test our assignment system, the model yields a recall of $M_{\text{TPR}} = 52.1\%$, using the same threshold. In Figure 4 we present the ROC curve for the Alert Model, calculated in months 4-8.

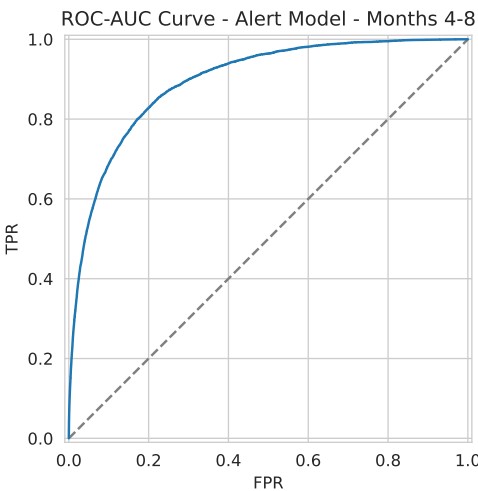

Figure 4: ROC-Curve - Alert Model - Months 4-8

## A.2 Determining $\lambda$ based on Neyman-Pearson criterion

In our task, the optimization goal is expressed by a Neyman-Pearson criterion, aiming to maximize recall subject to a fixed FPR of 5%. This criterion represents a trade-off between the cost of false positive $c_{\text{FP}}$ and the cost of false negative $c_{\text{FN}}$ errors. In bank account fraud prevention, the consequence of committing a false positive mistake, that is, rejecting a legitimate application, must be weighed against the cost of a false negative mistake, that is, accepting a fraudulent application. In Section 4.1, we state that our optimization goal is to obtain a set of predictions $\hat{y}$ that minimize the quantity

$$\frac{1}{N}\sum_{i=1}^{N}\left[\lambda\mathbb{I}[y_i = 0 \wedge \hat{y}_i = 1] + \mathbb{I}[y_i = 1 \wedge \hat{y}_i = 0]\right]. \tag{19}$$

However, in our task, we do not have access to the values of $c_{\text{FP}}$ and $c_{\text{FN}}$. To apply our misclassification cost re-weighting approach, we must then obtain the value $\lambda_t$ which is equivalent to the error cost trade-off enforced by the Neyman-Pearson criterion. This will allow us to set $c_{\text{FP}} = \lambda$ and $c_{\text{FN}} = 1$.

According to Elkan (2001), we can establish a relationship between the ideal threshold of a binary classifier and the misclassification costs. For a given instance, the ideal classification is the one that minimizes the expected loss. As such, the optimal prediction for any given instance $\boldsymbol{x_i}$ is 1 only if the expected cost of predicting 1 is less than, or equal to the expected cost of predicting 0, which is equivalent to:

$$(1-p)c_{\text{FP}} \leq pc_{\text{FN}} \tag{20}$$

Where $p = P(y = 1|\boldsymbol{x}_i)$, that is, the probability that $x_i$ belongs to the positive class. An estimation of the value of $p$ is given by our Alert Model, in the form of the model score output for a given instance, which is an estimate of the probability that $x_i$ belongs to class 1. In the case where the inequality is in fact an equality, then predicting either class is optimal. As such, the decision threshold $t$ for making optimal decisions leads us to a value for $\lambda$:

$$(1-t)c_{\text{FP}} = tc_{\text{FN}} \Leftrightarrow \lambda_t = \frac{t}{1-t} = 0.057 \tag{21}$$

As the optimal threshold $t$ for our Alert Model was chosen such that the Neyman-Pearson criterion is met, we now may plug the value of $t$ into this equation, obtaining the theoretical value $\lambda_t$ for our optimization goal. It has been shown by Sheng & Ling (2006) that this relationship between the cost-structure and the classifier's threshold does not always hold in practice. Secondly, the value of $\lambda$ obtained through this method depends on the classifier trained. A different classifier would yield another value for the optimal threshold according to the Neyman-Pearson criterion, which would lead to a different $\lambda$, despite the task being the same. However, as our aim is to test different cost structures, we will use this as the default value for $\lambda$, testing the cost-structures $\lambda \in \{\lambda_t/5, \lambda_t, 5\lambda_t\}$

## B    Classifier h

As detailed in Section 4.1, we model the classifier $h$ with a LightGBM (Ke et al., 2017) classifier. The model is trained on the alerts raised by the Alert Model, ranging from months four to six of the BAF dataset. The model is trained by minimizing the weighted log-loss as mentioned in section 3.1. The choice of hyperparameters is defined through Bayesian search (Akiba et al., 2019) on an extensive grid, for 300 total trials, with 200 start-up trials, with validation done on the seventh month. We also test several values for the initial probability estimate of the base predictor of the boosting model. This was done in order to test if introducing a bias towards predicting fraud can be beneficial to our model, as across all scenarios, false negatives incur a higher cost. Given instance-wise feature weights $c_i$, the default value $g_{\text{initial}}$ of the initial estimator's prediction is

$$g_d = logit\left(\frac{\sum_i c_i y_i}{\sum_i y_i}\right). \tag{22}$$

When training the model, we run the hyper-parameter search independently for $g_{\text{initial}} \in \{g_d, g_d + 0.2, g_d + 0.4, ..., g_d + 2\}$. The optimization objective is to minimize the weighted log-loss.

In a first series of experiments, we found that a low number of estimators and maximum depth resulted in the best results. As such, in our first thorough hyper-parameter search, we use the parameter space represented in Table 6.

Table 6: ML Model: LightGBM hyperparameter search space

| HYPERPARAMETER | VALUES OR INTERVAL | DIST. |
|---|---|---|
| boosting_type | "dart" | |
| enable_bundle | [False,True] | |
| n_estimators | [50,250] | Unif. |
| max_depth | [2,5] | Unif. |
| num_leaves | [100,1000] | Unif. |
| min_child_samples | [5,200] | Unif. |
| learning_rate | [0.001, 1] | Unif. |
| reg_alpha | [0.001, 2] | Unif. |
| reg_lambda | [0.001, 2] | Unif. |

In this set of experiments, across all scenarios, LGBM classifiers with a maximum tree depth of 2 achieved the best performance. As such, we conducted a second experiment, consisting of a total of 1700 trials, with 1500 startup trials, with the same hyperparameter space detailed in Table 6, but fixing the maximum depth parameter at 2.

## C   Synthetic Experts

### C.1   Expert Desiderata

**Feature and AI assistant dependence**   When a decision is made by an expert, it is assumed that they will base themselves on information related to the instance in question. Therefore, we expect experts to be dependent on an instance's features in order to make an informed decision.

However, in some real world deferral systems (Amarasinghe et al., 2022; De-Arteaga et al., 2020), the instance's features are accompanied by an AI model's score, representing the model's estimate of the probability that said instance belongs to the positive class. The aim of presenting the model score to an expert is to provide them with extra information, as well as possibly expediting the decision process. It has been shown that, in this scenario, expert's performance can be impacted by presenting the model's score when deferring a case to an expert (Amarasinghe et al., 2022; De-Arteaga et al., 2020; Levy et al., 2021). Note that, in this setting, a classifier trained for our task exists independently of the assignment system implemented. This approach is applicable to deferral systems such as the one proposed by Raghu et al. (2019a), who argue for the training of a separate human behaviour prediction model (Raghu et al., 2019a;b). In the L2D framework, however, the main classifier is trained jointly with the deferral system. As such, this framework allows users to generate experts with or without dependence on a separate ML classifier's score.

**AI assistance and algorithmic bias**   Should the generated experts use an AI assistant, we expect experts not to be in perfect agreement with the model, due to the assumption that humans and models have complementary strengths and weaknesses (De-Arteaga et al., 2020; Dellermann et al., 2019). As such, we would assume humans and AI perform better than one another in separate regions of the feature space, enabling an assignment system to obtain better performance than either the expert team or the model on their own. The degree of "model dependence", or "algorithmic bias" (Alon-Barkat & Busuioc, 2023),varies between humans, as measured by the model's impact on a human's performance (Jacobs et al., 2021; Inkpen et al., 2022). As such, our synthetic expert team may also exhibit varying levels of dependence on the model score.

**Varied Expert Performance**   In order for our team of experts to be realistic, it is important that these exhibit varying levels of overall performance. Experts within a field have been shown to have varying degrees of expertise, with some being outperformed by ML models (Goel et al., 2021; Gulshan et al., 2016). As such human decision processes can be expected to be varied even amongst a team of experts.

**Predictability and Consistency**   It is a common assumption that, when making a decision, experts follow an internal process based on the available information. However, it is also known that highly educated and trained individuals are still subject to flaws that are inherent to human decision making processes, one of these being inconsistency. When presented with similar cases, at different times, experts may perform drastically different decisions (Danziger et al., 2011; Grimstad & Jørgensen, 2007). Therefore we can expect a human's decision making process not to be entirely deterministic.

**Human Bias and Unfairness**   It is also important to consider the role that the assignment system can play in mitigating unfairness. If an expert can be determined to be particularly unfair with respect to a given protected attribute, the assignment system can learn not to defer certain cases to that expert. In order to test the fairness of the system as a whole, it is may be useful to create a team comprised of individuals with varying propensity for unfair decisions.

## C.2 Expert Parameter Sampling

As stated in section 4.1, we define the expert's probabilities of error, for a given instance, as a function of a pre-processed version of its features $\bar{\boldsymbol{x}}_i$ and the Alert Model's score $M(\boldsymbol{x}_i)$, given by

$$
\begin{cases}
\mathbb{P}(m_{j,i} = 1 | y_i = 0, \boldsymbol{x}_i) = \sigma\left(\beta_0 - \alpha \frac{\boldsymbol{w} \cdot \bar{\boldsymbol{x}}_i + w_M M(\boldsymbol{x}_i)}{\sqrt{\boldsymbol{w} \cdot \boldsymbol{w} + w_M^2}}\right) \\
\mathbb{P}(m_{j,i} = 0 | y_i = 1, \boldsymbol{x}_i) = \sigma\left(\beta_1 + \alpha \frac{\boldsymbol{w} \cdot \bar{\boldsymbol{x}}_i + w_M M(\boldsymbol{x}_i)}{\sqrt{\boldsymbol{w} \cdot \boldsymbol{w} + w_M^2}}\right),
\end{cases} \tag{23}
$$

Where $\sigma$ denotes a sigmoid function. Each expert's probabilities of the two types of error are parameterized by five parameters: $\beta_0, \beta_1, \alpha, \boldsymbol{w}$ and $w_M$. The sampling process of each parameter is done as follows:

**Feature Dependence Weights Generation** To define $\boldsymbol{w}$ for a given expert, we sample each component from a "Spike and Slab" prior (Mitchell & Beauchamp, 1988). A spike and slab prior is a generative model in which a random variable $u$ either attains some fixed value $v$, called the *spike*, or is drawn from another prior $p_{\text{slab}}$, called the *slab*. In our case, we set $v = 0$. That is, $u$ is either zero, or drawn from the slab density $N(0,1)$. To sample the values of $w_i$, we first sample a Bernoulli latent variable $Z \sim Ber(0.3)$ to select if $w_i$ is sampled from the *spike* or the *slab*. If $Z = 0$, $w_i$ attains the fixed value $v = 0$, if $Z = 1$, $w_i$ is drawn from the slab density $p_{\text{slab}}$. As such, the spike and slab prior induces sparsity unless $\theta = 1$, allowing for the generation of experts whose probabilities of error are swayed by a varying number of features. The distribution of $w_M \sim N(-2, 0.5)$ is defined separately to ensure all experts have some degree of model dependency. We also separately define $w_p \sim N(-1, 0.1)$, with $p$ representing the protected attribute, that is, the customer's age, so that all experts have a degree of unfairness in their simulated predictions, however, an exploration of fairness was not conducted in this work.

**Controlling Variability and Expert's consistency** While the weight vector controls the relative influence that each feature has on the probability of error, parameter $\alpha$, in turn, controls the global magnitude of this influence. For $\alpha = 0$, the probability of error would be identical for all instances. In turn, for very large $\alpha$, the probability would saturate at the extremes of the codomain of the sigmoid function, 0 or 1, resulting in a deterministic decision-making process. We chose $\alpha \sim N(4, 0.2)$ so that a wide variety of probability of errors exist throughout the feature space.

**Controlling Expert Performance** In any cost-sensitive binary classification task, the metric used to evaluate the performance of a decision-maker is the expected misclassification cost $\mathbb{E}[C]$.

For our setup to be realistic, we assume an expert's decisions must, on average, incur a lower cost than simply automatically rejecting all flagged transactions. Otherwise, assuming random assignment, having that expert in the human team would harm the performance of the system as a whole. We assume that $\mathbb{E}[C]_j$ must be, at most, 70% of the cost of rejecting all applications. We also desire to have a balanced distribution of expert performances. It is important that the humans in the system are not much worse on average than the classifier $h$, or else there could be no benefit to having humans in the decision-system as the cost of having the model predict on every instance is much lower than employing a team of human experts. The classifier $h$ can be trained before we even generate our human predictions, thus allowing us to know its average performance $\mathbb{E}[C]_h$ ahead of time. As such, we sample the target value of $\mathbb{E}[C]$ for each expert from $T_{\mathbb{E}[C]_j} \sim N(\mathbb{E}[C]_h, 0.2\mathbb{E}[C]_h)$, to ensure that the average expert performance is evenly distributed around the average performance of the classifier $h$. Should $T_{\mathbb{E}[C]_j}$ be larger than 70% of the cost of rejecting all alerts, the value is set to said value.

In Section 4.1 we demonstrate that the expected cost resulting from an expert $j$'s decisions is approximated by

$$
\begin{aligned}
\mathbb{E}[\mathcal{C}]_j \approx \frac{1}{N} \sum_{i=1}^{N} \big[ & \lambda \mathbb{P}(m_{j,i} = 1 | y_i = 0) \mathbb{P}(y_i = 0) \\
& + \mathbb{P}(m_{j,i} = 0 | y_i = 1) \mathbb{P}(y_i = 1) \big],
\end{aligned} \tag{24}
$$

where $\mathbb{P}(m_{j,i} = 1|y_i = 0)$ and $\mathbb{P}(m_{j,i} = 0|y_i = 1)$ are given by Equation 23. Assuming that the values of $\boldsymbol{w}, \alpha$ and $w_M$ are set, the value of $\mathbb{E}[C]_j$ will only be a function of $\beta_0$ and $\beta_1$. We must then be able to calculate values of $\beta_0$ and $\beta_1$ so that we obtain $\mathbb{E}[C]_j = T_{\mathbb{E}[C]_j}$. Let us simplify the above equation

$$
\begin{aligned}
T_{\mathbb{E}[C]_j} &= \lambda \mathbb{P}(y_i = 0)\frac{1}{N}\sum_{i=1}^{N}\left[\mathbb{P}(m_{j,i} = 1|y_i = 0)\right] \\
&+ \mathbb{P}(y_i = 1)\frac{1}{N}\sum_{i=1}^{N}\left[\mathbb{P}(m_{j,i} = 0|y_i = 1)\right], \\
&= \lambda(1 - \mathbb{P}(y_i = 1))\,\mathrm{FPR}_j + \mathbb{P}(y_i = 1)\,\mathrm{FNR}_j,
\end{aligned}
\tag{25}
$$

where, assuming all other parameters have been sampled, $\mathrm{FPR}_j = \mathrm{FPR}_j(\beta_0)$ and $\mathrm{FNR}_j = \mathrm{FNR}_j(\beta_1)$. There is a degree of freedom in this equation: if we choose the value of either $\beta_0$ or $\beta_1$, the other variable must attain a specific value so that the target $E[C]$ is achieved. Inverting the expression, we obtain:

$$
\mathrm{FPR}_j = \frac{T_{\mathbb{E}[C]_j}}{\lambda(1 - \mathbb{P}(y_i = 1))} - \frac{\mathbb{P}(y_i = 1)}{\lambda(1 - \mathbb{P}(y_i = 1))}\mathrm{FNR}_j,
\tag{26}
$$

Meaning that any pair of target values for $\mathrm{FPR}_j$ and $\mathrm{FNR}_j$ that obey the above equation will yield $T_{\mathbb{E}[C]_j}$. Having sampled $T_{\mathbb{E}[C]_j}$, we sample a random value of $T_{\mathrm{FNR}_j}$ such, and calculate $T_{\mathrm{FPR}_j}$ according to the above expression, ensuring both values belong to the interval $]0,1[$. We then calculate the value of $\beta_0$ and $\beta_1$ to obtain the desired FPR and FNR. From Equations 23, we know that

$$
\mathrm{FPR}_j(\beta_1) = \frac{1}{N}\sum_i \sigma\left(\beta_1 + \alpha\frac{\boldsymbol{w}.\boldsymbol{x}_i}{||\boldsymbol{w}||}\right).
\tag{27}
$$

Note that, for notational simplicity we set $w_M = 0$ but the result is similar when $w_M \neq 0$. It is then possible to show that the function $\mathrm{FPR}_j(\beta_1)$ is monotonically increasing,

$$
\frac{\partial \mathrm{FPR}_j}{\partial \beta_1} = \frac{1}{N}\sum_i \sigma\left(\beta_1 + \alpha\frac{\boldsymbol{w}.\boldsymbol{x}_i}{||\boldsymbol{w}||}\right)\left(1 - \sigma\left(\beta_1 + \alpha\frac{\boldsymbol{w}.\boldsymbol{x}_i}{||\boldsymbol{w}||}\right)\right) > 0 \text{ for } \beta \in \mathbb{R}.
\tag{28}
$$

Since the function is monotonically increasing, and in a bounded to the interval $]0,1[$, then, for any target false positive rate $T_{\mathrm{FPR}}$, then the following equation has an unique solution:

$$
\mathrm{FPR}_j(\beta_1) - T_{\mathrm{FPR}} = 0 \text{ for } T_{\mathrm{FPR}} \in ]0,1[.
\tag{29}
$$

A similar reasoning applies for $\beta_0$ and $T_{\mathrm{FNR}}$. Finally, we can control an expert's FPR and FNR by solving these equations for $\beta_1$ and $\beta_0$. To solve these equations, a partition of the dataset (month 7) is utilized to calculate the empirical value for each rate, meaning that deviations from the target performance metrics may occur due to temporal distribution shifts. Due to the monotonous nature of the function, and the uniqueness of the solution, we solve it through a bisection method (Burden & Faires, 1985).

**Feature Preprocessing** For our simulation of experts, the feature space is transformed as follows. Numeric features in $\mathbf{X}$ are transformed to quantile values, and shifted by $-0.5$, resulting in features with values in the $[-0.5, 0.5]$ interval. This ensures that the features impact the probability of error independently of their original scale. Categorical features are target-encoded, that is, encoded into non-negative integers by ascending order of target prevalence. These values are divided by the number of categories, so that they belong to the $[0, 1]$ interval, and shifted so that they have zero mean.

## C.3 Expert Properties

In this section we display and discuss some key properties of the expert teams generated for our experiments. As mentioned in Section 4.1, one team was generated per $\{a_r, \lambda\}$ pair, resulting in 6 distinct teams of 9 synthetic fraud analysts.

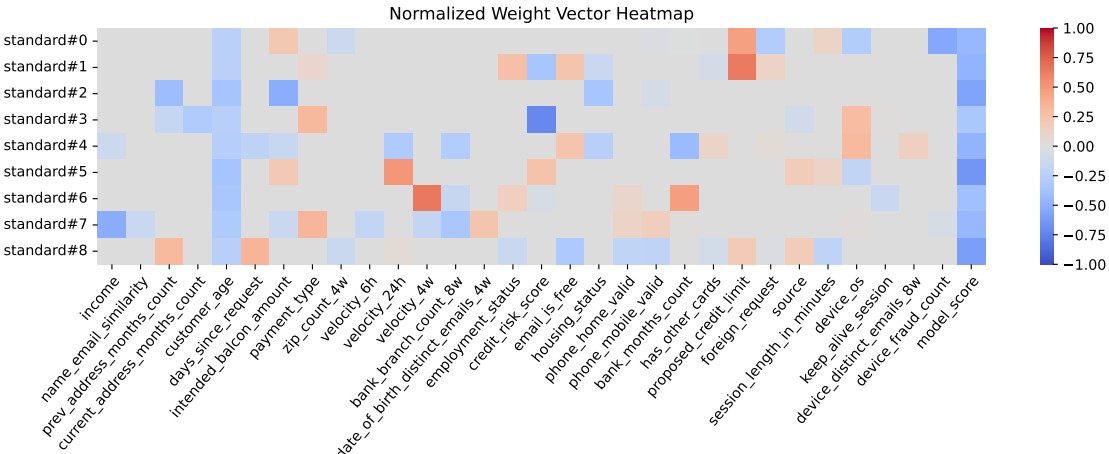

Figure 5: Weight Vector Heatmap for each Expert - Experts maintain feature weights across all testing scenarios.

**Feature Dependence**  To ensure that the deviations between testing scenarios are mainly due to the differences in alert rate $a_r$ and cost-structure, we generate all expert teams with the same feature weights so that their probabilities are similarly swayed by each instance. The normalized weight feature vector $w$ is represented in the heatmap in Figure 5, demonstrating the variety of decision processes within our team.

**Expert Performance**  In the previous section, we derive

$$\text{FPR}_j = \frac{\mathbb{E}[C]_j}{\lambda(1 - \mathbb{P}(y_i = 1))} - \frac{\mathbb{P}(y_i = 1)}{\lambda(1 - \mathbb{P}(y_i = 1))}\text{FNR}_j, \tag{30}$$

which allows us to establish a relationship between the expected misclassification cost for a given expert's decisions $\mathbb{E}[C]_j$ and their respective $FPR_j$ and $FNR_j$. Let us consider the "Full Rejection" approach, where we reject all the alerts. If we substitute the expected cost of rejecting all the alerts into Equation 30, we obtain the set of $FPR$ and $FNR$ values that result in the same cost in the test split. As such, in a plot where the x axis is the $FNR$ and the y axis is the $FPR$, all possible combinations of $FPR$ and $FNR$ leading to the same misclassification cost as rejecting all the alerts are represented by a single line. We exemplify this in figure 6, where the green area represents the combinations of $FPR$ and $FNR$ corresponding to a lower expected misclassification cost, while the red area represents a higher misclassification cost.

Note that $\mathbb{E}[C]_j$ only influences the intercept, meaning that lines parallel to the one represented in Figure 6 represent a set of (FPR,FNR) combinations with the same value of $\mathbb{E}[C]_j$. In Figure 7, we represent the distribution of $FPR$ and $FNR$ within the test split for all the generated expert teams. We also plot the performance of classifier $h$ (assuming 100% of training data availability) within the test set, demonstrating the relative performance of each decision-maker.

**Decision-Making Complementarity**  In this work, we assume that we stand to gain from combining the decision-making capabilities of various experts and a ML classifier. For this assumption to be true, we need to ensure that there are regions of the feature space where the probability of correctness of a given expert surpasses that of other experts, and vice versa. In Figure 8 we represent, on a heatmap, the fraction of total cases where the expert/classifier in the row is correct, while the expert/classifier in the column is incorrect. There is a significant number of instances, for all scenarios, where a given expert/classifier is a better choice than another decision-maker, meaning that this testbed is appropriate to test L2D methods.

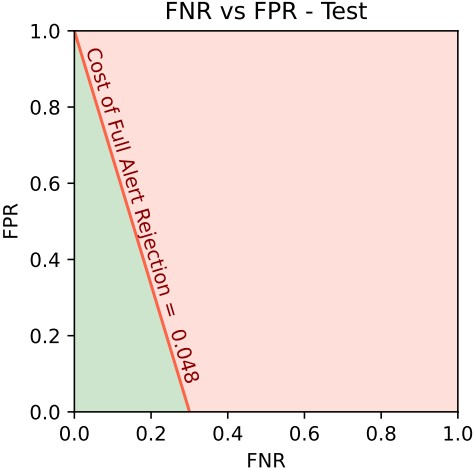

Figure 6: FPR vs FNR - Full rejection performance within the alerts. Red line represents combinations of (FPR,FNR) that result in the same cost as rejecting all instances

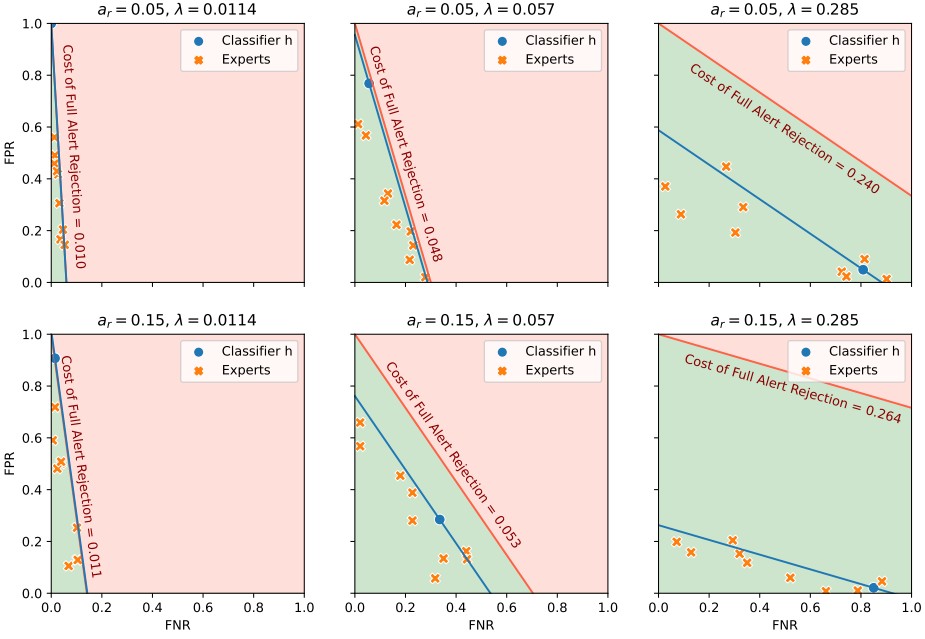

Figure 7: Expert and Classifier $h$ performance plots for each $\{a_r, \lambda\}$ pair

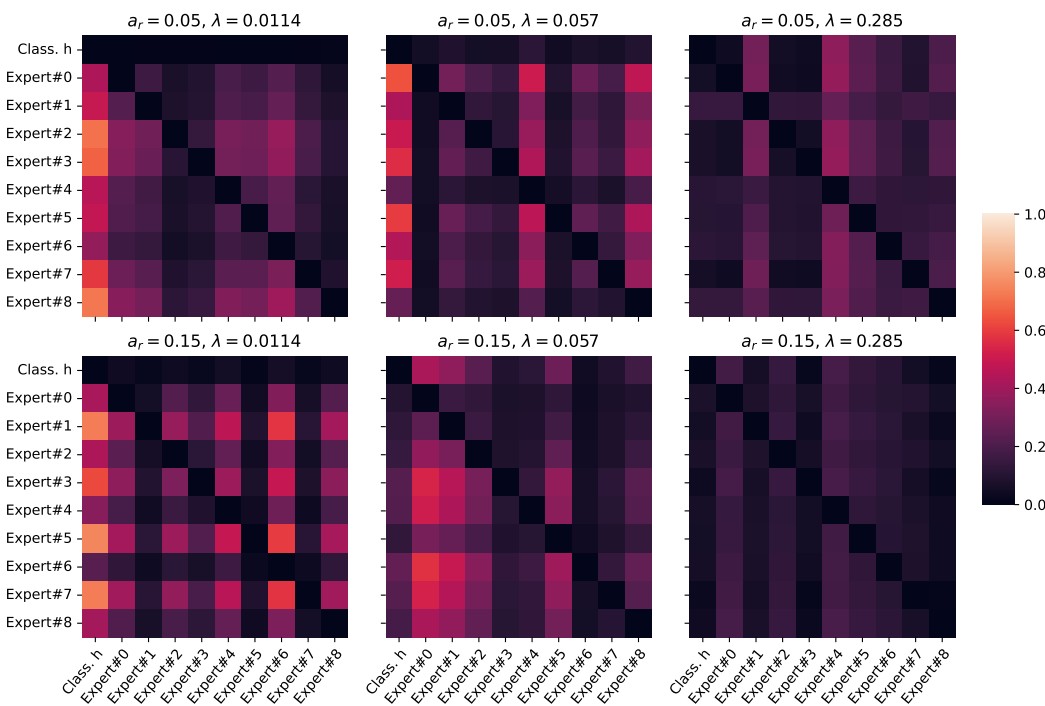

Figure 8: Fraction of instances in which ROW is correct and COLUMN is incorrect.

# D  OvA Classifiers and HEM

We model both the OvA Classifiers and the HEM with LightGBM (Ke et al., 2017) classifiers. These models are trained on the alerts raised by the Alert Model, and the corresponding expert predictions, ranging from months four to six of the BAF dataset. Both methods are trained by minimizing the weighted log-loss. The choice of hyperparameters is defined through Bayesian search (Akiba et al., 2019) on an extensive grid, for 120 total trials, with 100 start-up trials, with validation done on the seventh month. The hyperparameter search space is detailed in Table 7.

Table 7: LightGBM hyperparameter search space - OvA Classifiers and HEM

| HYPERPARAMETER | VALUES OR INTERVAL | DIST. |
|---|---|---|
| boosting_type | "dart" | |
| enable_bundle | [False,True] | |
| n_estimators | [50,250] | Unif. |
| max_depth | [2,20] | Unif. |
| num_leaves | [100,1000] | Log. |
| min_child_samples | [5,100] | Log. |
| learning_rate | [0.005, 0.5] | Log. |
| reg_alpha | [0.0001, 0.1] | Log. |
| reg_lambda | [0.0001, 0.1] | Log. |

