# OpenReview forum: "Cost-Sensitive Learning to Defer to Multiple Experts with Workload Constraints"
_TMLR — Accepted by TMLR_

### Review · Reviewer_zmkA · 2024-03-18

**Summary Of Contributions:**

Paper proposes the deferral under cost and capacity constraints framework extending the current works on learning to defer by
(i) dealing cost-sensitive scenarios, where type 1 and type 2 errors have different costs;
(ii) not requiring concurrent human predictions for every instance of the training
(iii) dealing with human work capacity constraints

**Audience:**

Yes

**Broader Impact Concerns:**

None.

**Claims And Evidence:**

Yes

**Requested Changes:**

Discuss
- the theoretical background. How this positive results can be supported with a theoretical justification?
- the challenges of the work especially for (i) and (ii) where quite standard approaches can be leveraged

**Strengths And Weaknesses:**

Strengths
- Idea is intriguing.
- Experimental result are promising.
Weaknesses
- it is not clear the theoretical background. How this positive results can be supported with a theoretical justification?
- it is not clear the challenges of the work especially for (i) and (ii) where quite standard approaches can be leveraged

---

> ### Author Response · Authors · 2024-04-25
> **Response to Reviewer zmkA**
>
> Response:
>
> Thank you for your thorough review. We will address your concerns in the following paragraphs.
>
> > Discuss
> > - the theoretical background. How this positive results can be supported with a theoretical justification?
>
> We believe that the theoretical foundation of our methodology is thoroughly detailed over the course of Section 3.
> Our approach to demonstrate the theoretical validity of our method hinges on the same principle as the works of Mozannar & Sontag and Verma et al., in that it focuses on proving that the obtained solution for the classifier-rejector pair agrees with the Bayes’ optimal classifier, as derived by Verma et al., in the multi-expert L2D setting. The mathematical demonstrations needed to support this conclusion are detailed in Section 3.1.
>
> Additionally, in Section 3.3, we describe the drawbacks of considering the pointwise optimal solutions as the best set of assignments, since in a real-world scenario, where capacity constraints exist, a better set of assignments can be achieved by using a batch-wise constraint programming approach, thus globally minimizing the probability of error.
>
> Finally, in the paragraph “Rejector Training”, in Section 3.1, we describe how the chosen architecture (a single model for all experts) may prove beneficial, as, in this way, human behavior is modeled jointly. Note that this hypothesis is in direct contradiction with the authors of the One vs. All Parameterization, who state that the lack of dependence across expert correctness scoring functions is expected to result in better calibration in practice (see Section 4.2 in [source]). In our experimental results (see Figure 2), we demonstrate that no approach consistently leads to a reduction in the expected calibration error, as opposed to Verma et al.s assumption.
>
> > Discuss
> > -   the challenges of the work especially for (i) and (ii) where quite standard approaches can be leveraged
>
> **L2D under cost-sensitive scenarios:** In this work, we consider the challenges imposed by cost-sensitive scenarios, mainly the high class imbalance inherent to such use-cases. We do agree that standard approaches can be employed to deal with these hurdles, and in our work we use an instance re-weighting approach, a commonly used method.
>
> Although standard approaches can be leveraged to tackle these issues, we find that it is crucial to stress the importance of studying these system’s behavior under such conditions, as most high-stakes domains (e.g. Diagnostic Medicine, Financial Fraud), these challenges are present.
>
> Unfortunately, previous research neglects to consider these scenarios and the impacts they may have in the system’s performance. Note how, in section 5.2, we demonstrate how the ECE is highly variable across scenarios where only the cost structure is changed, even changing the ranking of the evaluated methods in terms of calibration of expert correctness.
>
> **L2D without concurrent human predictions for every instance in training:** To tackle this issue, standard approaches such as imputing the missing predictions, as proposed by [1], or active learning, as proposed by [2] could potentially be applied.
>
> Note, however, that these approaches were only developed for the single expert use case, and their generalization to the multi-expert scenario is not trivial. Furthermore, employing these would incur further developmental costs, or require querying experts. Our proposed solution to this issue - the decoupling of scoring functions in the OvA approach, and the joint modeling of expert behavior in the DeCCaF approach - is immediately applicable to scenarios where only one prediction per expert is available per instance.
>
> *Sources*:
>
> [1] Hemmer, P., Thede, L., Vössing, M., Jakubik, J., & Kühl, N. (2023, June). Learning to defer with limited expert predictions. In Proceedings of the AAAI Conference on Artificial Intelligence (Vol. 37, No. 5, pp. 6002-6011).
> [2] Charusaie, M. A., Mozannar, H., Sontag, D., & Samadi, S. (2022, June). Sample efficient learning of predictors that complement humans. In International Conference on Machine Learning (pp. 2972-3005). PMLR.

---

### Review · Reviewer_N8kV · 2024-04-17

**Summary Of Contributions:**

In the learning to defer setting, a classifier is allowed to defer to an expert on samples it deems to be difficult. Recently, Verma et al. (2023) provide a learning algorithm for a variant of this setting, where the classifier has multiple experts it can defer to. The present paper builds on this prior work and considers applications where there is a capacity constraint on each expert.

The proposed solution assumes access to a dataset labeled with expert predictions, and requires learning a classifier and a separate model for predicting the accuracy of each expert. During inference on a batch of sample, the authors propose solving a *constraint program* to assign experts to instances where the classifier ought to defer. This approach additionally accommodates asymmetric costs for misclassification errors, and does not require each training sample to be annotated with predictions from all the experts.

On a bank fraud detection, the authors demonstrate the efficacy of their approach over that of Verma et al. (which they alter using a heuristic to accommodate capacity constraints).

**Audience:**

Yes

**Claims And Evidence:**

Yes

**Requested Changes:**

Some adjustments the authors could do to improve the quality of the paper include:
- Present experimental comparisons with more datasets (preferably with those used in prior papers on L2D), and with different models (e.g. neural networks). Explore multi-class datasets too.
- Evaluate the efficacy of their approach across a range of capacity constraints. In settings with a single expert, one could vary the capacity for the single expert, and plot deferral curves that trace accuracy as a function of capacity (see e.g. Verma & Nalisnick). With multiple experts, such a clean evaluation may be difficult. The authors could find a single parameter to vary, such as the sum of capacities or the skewness in capacity amongst experts, and trace out the resulting accuracies as a function of this parameter. Deferral curves like this would allow us to visually compare different methods across a range of capacity constraints.
- Fix the notational issues with the constraint program formulation in (7)

Having said this, given that my main complaint is about novelty, I am not sure if making the experimental evaluation more comprehensive alone will suffice for acceptance.

Here's a related direction that would make the paper more compelling. I do not expect the authors to pursue this idea for this submission cycle:
- It would be interesting to incorporate the capacity constraints into the training objective instead of at inference time. You could e.g. consider the OvA objective of Verma et al. (equation in Section 4.2). Using a Lagrange formulation, the capacity constraints could then be translated into "weights" on the individual expert loss terms. The challenge would then be to choose these weights in a efficient manner, so as to satisfy the capacity constraints.

**Strengths And Weaknesses:**

**Strengths:**
- The problem of learning to defer has huge practical relevance, and deferral to multiple experts with budget constraints is a natural and realistic setting to look at.
- Unlike the setting with a single expert (Mozannar & Sontag, 2020), its not immediate how one may modify existing approaches to incorporate budget constraints on multiple experts.
- The idea of solving a constraint program on estimates of classifier and expert accuracies is a simple and effective idea; in fact such post-hoc methods have proven to be effective in other learning to reject/defer settings.

**Weaknesses:**
- The main contribution is setting up a post-hoc constraint program to assign experts to samples. I do not consider the inclusion of asymmetric misclassification costs or allowing for limited expert annotations to be significant contributions. Given this, the paper's novelty seems limited.
- The experimental evaluation is also limited, in that all the comparisons are done on a *single* binary classification dataset.
- The formulation of the constraint program in (7) is confusing to parse through. The objective in (7) simplifies to:
$$
\sum_{i=1}^n \sum_{j=1}^J \mathbb{P}(correct | x_i) \cdot A_{ij} = \sum_{i=1}^n \mathbb{P}(correct | x_i) \cdot \sum_{j=1}^J  A_{ij} = \sum_{i=1}^n \mathbb{P}(correct | x_i),
$$
with the optimization variables $A_{ij}$ summing up to 1 and vanishing from the objective. Moreover, the definition of $\mathbb{P}(correct | x_i)$ is confusing as it uses an assignment function $a(x)$ (see (6)) which is never explicitly included in the optimization problem in (7).
- The experiments are all on LightGBM models, due to which the authors are not able to jointly train the classifier $h$ and human expert model with shared parameters. In contrast, Verma et al. model both $h$ and the defer logits using the same neural network backbone, but different prediction heads. This allows for joint training with shared parameters; consequently, one could expect the learned logits to generalize better.

**Typos / Presentation**
- Page 5: $g_\perp: \mathcal{X} \cup \\\{1,\ldots, J\\\} \rightarrow \mathbb{R}$. Change to: $g_\perp: \mathcal{X} \times \\\{1,\ldots, J\\\} \rightarrow \mathbb{R}$
- Page 5: $\psi^{-1}(gg(x))$. Change to $g(x)$
- Sec 3.4: This section was difficult to follow. Specifically, it wasn't clear how it relates to the content in the previous sections. For example, how would the inclusion of misclassification costs change the Bayes-optimal solution in (2). I think the authors are trying to say that the inclusion of costs is equivalent to computing the solution in (2) with an altered distribution. However, this wasn't entirely clear from the discussion in Sec 3.4.

---

> ### Author Response · Authors · 2024-04-25
> **Response to Reviewer N8kV #1**
>
> Thank you for your thorough review. We have updated our paper to include your recommendations, and will address your concerns individually in the following paragraphs. As your primary complaint concerns a lack of novelty, and changes to the experimental setup may be unfeasible in this submission cycle, we will focus on responding to the criticisms regarding our method’s contributions.
>
> > The main contribution is setting up a post-hoc constraint program to assign experts to samples. I do not consider the inclusion of asymmetric misclassification costs or allowing for limited expert annotations to be significant contributions. Given this, the paper’s novelty seems limited.
>
> The inclusion of asymmetric misclassification costs is of paramount importance in the field of L2D, since they are a reality in most real-world high-stakes scenarios in which Human-AI Collaboration systems could be employed (e.g. diagnostic medicine, financial fraud prevention). In this work, we consider the challenges imposed by cost-sensitive scenarios, mainly the high class imbalance inherent to such use-cases. We agree that standard approaches can be employed to deal with these hurdles, and in our work we use an instance re-weighting approach, a commonly used method.
>
> Although standard approaches can be leveraged to tackle these issues, previous research neglects to consider these scenarios and the impacts they may have in the system’s performance. Note how, in Section 5.2, we demonstrate how the ECE is highly variable across scenarios where the cost structure is changed, being significantly higher for the most imbalanced cost-structure. Furthermore, we observed that changes in the cost structure resulted in different rankings of the evaluated methods in terms of calibration of expert correctness (see Figure 2). We believe that this is a novel and important conclusion that should be taken into account when designing L2D systems.
>
> The issue of limited expert annotations is also of great importance to the applicability of L2D in real-world systems. Again, standard approaches such as imputing the missing predictions, as proposed by [1], or active learning, as proposed by [2] could potentially be applied. Note, however, that these approaches were only developed for the single expert use case, and their generalization to the multi-expert scenario is not trivial. A direct adaptation to the multi-expert scenario would result in both of these approaches still requiring the existence of every expert’s annotations for every data point in training. Furthermore, employing these would incur further developmental costs, and, in the active learning case, require querying experts.
>
> Our proposed solution to this issue - the decoupling of scoring functions in the OvA approach, and the joint modeling of expert behavior in the DeCCaF approach - is immediately applicable to scenarios where only one expert’s annotation is available per instance, a much less restrictive data requirement.
>
> > The formulation of the constraint program in (7) is confusing to parse through. The objective in (7) simplifies to:
> $\sum_{i=1}^n \sum_{j=1}^J \mathbb{P}(correct | x_i) \cdot A_{ij} = \sum_{i=1}^n \mathbb{P}(correct | x_i) \cdot \sum_{j=1}^J  A_{ij} = \sum_{i=1}^n \mathbb{P}(correct | x_i),$
>
> Thank you for raising this concern. The formulation in (7) is not correct as the probability of correctness must also be dependent on the assignment decision. Furthermore, the dimensionality of the matrix of assignments $A$  is not correct, as there are a total of $J+2$ assignment options: the classifier predicts 0 or 1, or the instance is deferred to one of J experts. The current formulation accidentally disregards the possibility of automatic classification.
>
> We have corrected this notational error by changing $\hat{\mathbb{P}}(correct|\mathbf{x}_i)$ to  $\hat{\mathbb{P}}(correct|\mathbf{x}_i, a_i)$, as the probability of correctness is specific to both the instance and the possible assignment decision $a_i \in \\{ 1, ..., J+2 \\} $, where $a_i=y+1$ with $y\in \cal{Y}$ , is an automatic prediction of class $y$ for instance $i$, whereas $a_i=j+2$ denotes the decision to defer instance $i$ to the $j$th expert. The new formulation in (7) is therefore:
>
> $\sum_{i=1}^{n_b}  \sum_{a_i=1}^{J+2} \hat{\mathbb{P}} (correct | \mathbf{x}_i ,  a_i ) A _{i, a _i}$.

---

> ### Author Response · Authors · 2024-04-25
> **Response to Reviewer N8kV #2**
>
> > Moreover, the definition of $\mathbb{P}(correct | x_i)$ is confusing as it uses an assignment function $a(x)$ (see (6)) which is never explicitly included in the optimization problem in (7).
>
> In the original version of the paper, the definition of an assignment function $a(x)$ is an auxilliary step to define the probability of expert/classifier correctness $\hat{\mathbb{P}}(correct|\mathbf{x}_i)$. We agree that defining a new function is confusing, and we have changed the definition according to the previous paragraph, simplifying our notation to consider the possible assigner decisions $a_i $ for a given instance $i$, instead of an hypothetical assignment function.
>
> > The experiments are all on LightGBM models, due to which the authors are not able to jointly train the classifier $h$ and human expert model with shared parameters. In contrast, Verma et al. model both $h$ and the defer logits using the same neural network backbone, but different prediction heads. This allows for joint training with shared parameters; consequently, one could expect the learned logits to generalize better.
>
> In our implementation of the OvA approach we use separate functions for each of the scoring functions due to the fact that our data is of the form $S = \\{ \mathbf{x} _{i} , y_i , m _{j,i} , c_i \\} ^{N} _{i=1}$ that is, only one expert prediction associated with each instance.
>
> In this use-case, the surrogate loss proposed by Verma et al. can not be used directly. As such, our adaptation consists of separating these scoring functions into completely separate classifiers which can be trained in the subset of data pertaining to each expert’s predictions. Note that despite sharing the same neural network backbone, Verma et al. also consider the scoring functions to be independent of one another, as there is no backpropagation across them. In fact, the authors state that the independence of the scoring functions is a positive attribute of their approach, arguing that it should produce better calibration in practice (see Section 4.2 of [3]). As such, we feel that our implementation of the OvA approach in this use case is adequate.
>
> >  Typos/Presentation
> >
> > - Page 5: $g_\perp: \mathcal{X} \cup \{1,\ldots, J\} \rightarrow \mathbb{R}$. Change to $g_\perp: \mathcal{X} \times \{1,\ldots, J\} \rightarrow \mathbb{R}$
> > - Page 5: $\psi^{-1}(gg(x))$. Change to $g(x)$.
> > - Sec 3.4: This section was difficult to follow. Specifically, it wasn’t clear how it relates to the content in the previous sections. For example, how would the inclusion of misclassification costs change the Bayes-optimal solution in (2). I think the authors are trying to say that the inclusion of costs is equivalent to computing the solution in (2) with an altered distribution. However, this wasn’t entirely clear from the discussion in Sec 3.4.
>
> Thank you for pointing out these typographical errors. They are corrected in the updated version of the paper.
>
> Regarding your comments about Section 3.4: this section intends to clarify how the training and optimization processes detailed in sections 3.1 and 3.3, respectively, would be conducted in a cost sensitive scenario. In this section we demonstrate that, to apply our method to a cost sensitive scenario, we would simply re-weight instances during training. This is analogous to applying the 0-1 Loss minimization under a re-weighted distribution (e.g. The probability correctness estimates would be calibrated under the altered distribution, but not under the original one). We have rephrased the beginning of this Section to clarify that this is the approach used during training.

---

> ### Author Response · Authors · 2024-04-25
> **Response to Reviewer N8kV #3**
>
> > Here’s a related direction that would make the paper more compelling. I do not expect the authors to pursue this idea for this submission cycle:
> >
> > It would be interesting to incorporate the capacity constraints into the training objective instead of at inference time. You could e.g. consider the OvA objective of Verma et al. (equation in Section 4.2). Using a Lagrange formulation, the capacity constraints could then be translated into “weights” on the individual expert loss terms. The challenge would then be to choose these weights in a efficient manner, so as to satisfy the capacity constraints.
>
> This is an interesting suggestion, and similar approaches have been proposed for the single-expert case. However, while such a formulation could allow for regularization of deferral, across experts (e.g. one expert can predict in 100 instances per batch while another can predict in 70), it would not be adequate for a real-world system, where the availability of experts may vary drastically across batches. In a real-world system, different teams may work at different schedules, which would result in experts being available in some batches while absent in others. The existence of changes in expert availability within real-world HAIC scenarios means that our system must be able to accommodate constraints which are defined at inference-time, and not during training.
>
>
>
> *Sources*:
>
> [1] Hemmer, P., Thede, L., Vössing, M., Jakubik, J., & Kühl, N. (2023, June). Learning to defer with limited expert predictions. In Proceedings of the AAAI Conference on Artificial Intelligence (Vol. 37, No. 5, pp. 6002-6011).
>
> [2] Charusaie, M. A., Mozannar, H., Sontag, D., & Samadi, S. (2022, June). Sample efficient learning of predictors that complement humans. In International Conference on Machine Learning (pp. 2972-3005). PMLR.
>
> [3] Verma, R., Barrejón, D., & Nalisnick, E. (2023, April). Learning to defer to multiple experts: Consistent surrogate losses, confidence calibration, and conformal ensembles. In International Conference on Artificial Intelligence and Statistics (pp. 11415-11434). PMLR.

---

> > ### Comment · Reviewer_N8kV · 2024-05-28
> > **Re:Response to Reviewer N8kV**
> >
> > I thank the authors for the detailed rebuttal and for revising the manuscript. The authors do address some of the concerns I had.
> >
> > > Re: asymmetric costs and limited experts annotations
> >
> > I totally understand that these are important practical issues and am glad that the paper highlights them. My comment about limited novelty had to do with the specific techniques used to address these issues. I could have been a bit more clearer about this in my initial review.
> >
> > > Re: correction to formulation
> >
> > Thanks for the corrections made.
> >
> > > Re: sharing of parameters
> >
> > Thanks for the pointer to Verma et al.'s note that they expect independent scorers to result in better calibration. In general, I would expect allowing for some sharing of parameters between scorers to be useful, particularly with large neural networks.
> >
> > As for the surrogate loss of Verma et al. not being directly applicable when there is only one expert prediction per sample: I agree with this. Your adoption of their loss to settings with missing expert annotations makes sense. However, even in your adoption of the OvA loss, one could jointly train the scorers to optimize the sum of binary CE losses on each scorer, and thus allow for some parameter sharing amongst the individual models.
> >
> > In any case, the lack of parameter sharing was a minor observation and not something I hold against the present paper.
> >
> > Overall, I am still a bit worried about the novelty being limited. I look forward to discussing this further with the other reviewers and the AE. Thanks again for the detailed explanations.

---

### Review · Reviewer_UTzZ · 2024-04-18

**Summary Of Contributions:**

The paper is based on Learning to Defer (L2D), which aims to enhance human-AI collaboration systems by deferring decisions to humans when they are more likely to be correct than a machine learning (ML) classifier. Existing L2D research overlooks practical challenges such as cost-sensitive scenarios, concurrent human predictions, and human work capacity constraints.

To address these challenges, the paper proposes the deferral under the cost and capacity constraints framework (DeCCaF). DeCCaF employs supervised learning to model human error probabilities with less restrictive data requirements and uses constraint programming to minimize error costs while considering workload limitations. The authors test DeCCaF in cost-sensitive fraud detection scenarios with teams of synthetic fraud analysts and an ML classifier.

The paper provides a new benchmark of complex, feature-dependent synthetic expert decisions in a realistic financial fraud detection scenario. Through empirical testing in various cost-sensitive fraud detection scenarios, the paper demonstrates that DeCCaF outperforms existing L2D baselines in terms of average misclassification costs.

**Audience:**

Yes

**Broader Impact Concerns:**

None.

**Claims And Evidence:**

Yes

**Requested Changes:**

Requested changes.

1. Please add an algorithm block, detailing the step-by-step procedure.

2. Clear definitions of capacity constraints, $H$, $b$, $A$, etc. with some examples would be helpful. Please have a summary of the notations in the paper.


3. Minor notation nitpick, commonly the optimal solution ( Bayes optimal solutions) are denoted as $h^*, g^*,$ etc. and the solutions of the empirical problem are denoted as $\hat{h}$, $\hat{g}$. The paper uses $\hat{P}$ but it is defined on $h^*, g^*$ which are the empirical solutions in the paper. This leads to confusion.


4. The rejector function $r: \mathcal{X} \to \\{0,1,\ldots, J\\}$, is rather an assignment/deferral function. Could you please clarify or update it.

**Strengths And Weaknesses:**

### Strengths

The paper presents DeCCaF, a novel framework designed to defer decisions to humans within the cost and capacity constraints, addressing practical challenges in the field of Learning to Defer (L2D). DeCCaF utilizes supervised learning to model human error probabilities using one expert prediction per instance. It employs constraint programming to minimize error costs while considering workload limitations. Empirical evaluation with nine simulated fraud analysts showcases DeCCaF's superiority over existing L2D methods, resulting in reduced misclassification costs across diverse scenarios.


### Weaknesses/Questions

The method requires data labels $m_{j,i}$ for each data point $i$ and each expert $j$. It may be impractical to get labels for all data points from each expert. The method has multiple steps and it is not clear how they are executed in sequence. My current understanding is that first $h^*$ is learned then $r^*$ and then the allocation matrix $A$. Shouldn’t the allocation be taken care of while learning $r^*$ itself? Do the allocation, accuracies, and costs get revised after each instance, e.g. the available capacity of an expert will be reduced after making a prediction?  What are the trade-offs between the overall performance and the cost, capacity, and accuracies of the experts, and classifiers?

---

> ### Author Response · Authors · 2024-04-25
> **Response to Reviewer UTzZ**
>
> Thank you for the thorough review. Over the next paragraphs we will address the weaknesses and questions raised, as well as the requested changes.
>
> >The method requires data labels for each data point and each expert. It may be impractical to get labels for all data points from each expert.
>
> We agree that it is impractical to demand labels for all data points from each expert, and we criticize this requirement in the works of Mozannar & Sontag as well as Verma et al., both works which assume the existence of $m_{i,j}$ for $i \in \{1,...,N\}$ and $j \in \{1,..,J\}$. This is, however, not applicable to our work, where the training set is assumed to be of the form  $S = \\{ \mathbf{x} _{i} , y_i , m _{j,i} , c_i \\} ^{N} _{i=1}$ , that is, only containing one expert prediction per training instance. See also section 3.1, in particular the paragraph "Rejector Training", where we further justify our model architecture based on the training set S, which assumes the availability of a single expert prediction per instance.
>
> > The method has multiple steps and it is not clear how they are executed in sequence. My current understanding is that first $h^*$ is learned then $r^*$  and then the allocation matrix $A$. Shouldn’t the allocation be taken care of while learning $r^*$ itself?
>
> $h^*$ and $r^*$ are indeed learned separately, with the primary objective of producing calibrated estimates of the probabilities of correctness $\hat{\mathbb{P}}(\text{correct} \lvert \mathbf{x}_i, a_i)$ for each possible assignment decision $a_i$ (see the response to Reviewer N8kV for clarification regarding this notation).
>
> The allocation matrix $A$ is not learned, but calculated at inference time, by solving a constraint programming problem (see equation (7) in the paper) subject to the experts' capacity constraints at inference time. Thus, having trained $h^*$ and $r^*$, the matrix A is calculated for each batch of instances defined by $b$, while respecting the capacity constraints defined in $H$.
>
> We have clarified this step in the algorithm as being detached from the training process (see new version of Section 3.3). Finally, we believe the inclusion of an algorithm block, as requested, clarifies this issue.
>
> > Do the allocation, accuracies, and costs get revised after each instance, e.g. the available capacity of an expert will be reduced after making a prediction?
>
> We are not sure what is meant by “accuracies”. The costs are defined as part of the use-case and are therefore assumed constant throughout the deployment of the system. The allocation is done batch-wise at inference time, based on the capacity constraints of the experts (see previous paragraph). The available capacity of an expert does not need to be reduced after making a prediction, as their capacity is taken into account when solving the optimization problem detailed in equation (7).
>
> We believe the inclusion of the algorithm block and the response to the previous observation further provide clarity in this issue.

---

> > ### Author Response · Authors · 2024-04-25
> > **Response to Reviewer UTzZ #2**
> >
> > > Clear definitions of capacity constraints, H, b, A, etc. with some examples would be helpful. Please have a summary of the notations in the paper.
> >
> > Thank you for this suggestion, we have included a summary of the notations in the revised version of the paper.
> >
> > >Minor notation nitpick, commonly the optimal solution ( Bayes optimal solutions) are denoted as $h^*$, $g^*$ etc. and the solutions of the empirical problem are denoted as $\hat{h}$, $\hat{g}$. The paper uses $\hat{P}$ but it is defined on $h^*$, $g^*$ which are the empirical solutions in the paper. This leads to confusion.
> >
> > Thank you for raising this concern. To make the notation more easily comprehensible, we have introduced this change in the revised version of the paper.
> > > The rejector function $r : \mathcal{X} \rightarrow \{0,1,...,J\}$, is rather an assignment/deferral function. Could you please clarify or update it.
> >
> > We chose to name $r$ the rejector function due to the commonly used rejector-classifier formulation considered in the L2D and Rejection Learning research community, see [1],[2],[3]. In the multi-expert scenario, the function responsible for selecting the expert to which the instance is assigned is still referred to as the rejector in the work of Verma et. al. [4]. For consistency, we decided to maintain the nomenclature.
> >
> > *Sources*:
> > [1] Mozannar, H., & Sontag, D. (2020, November). Consistent estimators for learning to defer to an expert. In International Conference on Machine Learning (pp. 7076-7087). PMLR.
> > [2] Cortes, C., DeSalvo, G., & Mohri, M. (2016). Learning with rejection. In Algorithmic Learning Theory: 27th International Conference, ALT 2016, Bari, Italy, October 19-21, 2016, Proceedings 27 (pp. 67-82). Springer International Publishing.
> > [3] Verma, R., & Nalisnick, E. (2022, June). Calibrated learning to defer with one-vs-all classifiers. In International Conference on Machine Learning (pp. 22184-22202). PMLR.
> > [4] Verma, R., Barrejón, D., & Nalisnick, E. (2023, April). Learning to defer to multiple experts: Consistent surrogate losses, confidence calibration, and conformal ensembles. In International Conference on Artificial Intelligence and Statistics (pp. 11415-11434). PMLR.

---

### Author Response · Authors · 2024-07-13
**Submission of Final Camera-Ready Version**

We once again thank the Reviewers and Action Editor for their thorough work in the evaluation and improvement of our work. We have now submitted the final camera-ready version, where we introduce the Action Editor suggestions apart from new experiments, as justified in the previous message.

You will also find the link to the relevant code and a video presentation of our work in the camera-ready submission. The repository may take a few work days to become public, as it is currently going through security clearances within the company. It is currently available in its anonymized version here: https://anonymous.4open.science/r/deccaf-1245/README.md

Thank you very much for your support and for the acceptance of our work.

Sincerely,

The authors

---

### Decision · Action_Editor_W7Yf · 2024-06-16

**Recommendation:** Accept with minor revision

**Comment:**

Suggested changes for final version:
- Figure 1, consider moving to top of page 2 rather than having inline in Abstract
- notation, is there a reason for $y$ to not be italicised?
- pg 4, typo: "depend on ithe features"
- Equation 1, make the LHS $L_{01}(h, r)$ to specify the variables of interest
- Equation 2 and elsewhere, align "if" and "otherwise"
- Equation 7, split the constraints onto two separate lines
- Equation 7, please comment on the computational complexity of the suggested CP-SAT solver.
- pg 7, is there a need to introduce a new symbol $k$ in defining the cost-sensitive loss?
- Algorithm 1, suggested to split the Training and Inference algorithms into two
- Algorithm 1, not ideal to overload "Train" for fitting of both $\hat{g}$ and $\hat{g}_{\perp}$. More precise equation references in Section 3.1 are advisable.
- Algorithm 1, specify the size of $\mathbf{H}$
- Algorithm 1, give a comment specifying the meaning of $\mathbf{A}$
- Algorithm 1, line 9 and following, $i \leftarrow \\{ 1, \ldots, J + 2 \\}$
- citations to the MoE literature are potentially relevant, as these also consider expert load, albeit for slightly different reasons. e.g., Zhou et al., Mixture-of-Experts with Expert Choice Routing, NeurIPS 2022.
- there are more relevant citations for multi-expert deferral, e.g., Mao et al., Two-Stage Learning to Defer with Multiple Experts, NeurIPS 2023.
- use $c_{\rm FP}$ instead of $c_{FP}$
- Equation 12, use of braces is not necessary

- if possible to obtain results on even a single task that was used in prior work on learning to defer to multiple experts, it would strengthen the paper.

**Audience:**

All reviewers were in agreement that the problem considered -- deferring to multiple experts, with a budget on the capacity of each expert -- is of interest to the community. The problem of learning to defer to an expert has received good attention in the literature in recent years, and is expected to be of growing importance given its applications to problems such as human-AI collaboration.

**Claims And Evidence:**

The paper's central claim is that it provides a new technique for addressing the problem of deferring to multiple experts, where there is a capacity constraint on each expert. It is claimed that this technique is superior to naive baselines that are based on current techniques, which do not involve such a capacity constraint.

In the initial set of reviews, there were several suggestions aimed at improving the precision and clarity of the paper. These were largely incorporated in the revised version of the paper. Following this, two reviewers were weakly positive about the paper's contributions being convincingly demonstrated. One reviewer maintained a more qualified view, particularly regarding the experimental results, which are on a single dataset with binary labels. Even on the single dataset, it was suggested that a deeper analysis of the behaviour of the methods as the capacities are varied would be of interest. (The reviewer also raised separate points on novelty, which are valid but do not influence this specific criterion.)

We agree that it would indeed be ideal to have results on a broader range of settings, including those with multiple possible classes, and different model families. At the same time, we do not have strong reasons to believe that the results would not generalise to such settings. Thus, on balance we believe the results as presented do evince the paper's central claims.

---

> ### Author Response · Authors · 2024-06-25
> **Response to Final Decision and Suggested Changes**
>
> We thank the Reviewers and Action Editor for taking the time to review our work and for the thoughtful and constructive feedback that helped us to revise our paper according to their recommendations and criticisms. All suggested changes regarding typos, equations, notation, the inclusion of new citations, and improvement of the algorithm block will be implemented in the revised version.
>
> Though we agree that the paper would benefit from having results on a broader range of datasets with different model architectures, we think a new set of experiments covering these extra datasets should be the target of further work, for the reasons described below.
>
> First, note that our paper criticizes previous L2D testing [1][2][3] for employing overly simplistic simulated human behavior. To make our test setting more realistic, we propose a novel benchmark of complex and feature dependent experts, which we believe significantly increases the difficulty of the task. However, this synthetic decision generation method was designed with tabular data in mind, which has so far been neglected in L2D literature. To the best of our knowledge, previous L2D testing is done mostly in computer vision datasets, most commonly CIFAR-10. As such, we do not believe that our results would be comparable to those obtained if we replicate previous L2D testing, and would not complement the central claims of our paper.
>
> Second, the adaptation of our algorithm to neural networks is not as straightforward as just rerunning our experiments, as the change in model architecture would raise several questions (i.e. Should we train classifier h separately for (potentially) better calibration, as argued by Verma et al. [4], or should we allow weight sharing for potential complementarity between the classifier and expert predictors?), significantly increasing the complexity of running such experiments and analyzing their results.
>
> Concluding, as we believe the impact of the choice of ML algorithm and model architecture on L2D performance merits an in-depth study on a wide range of datasets, we will include this in our "Conclusions and Future Work" Section.
>
> [1] Hussein Mozannar, Hunter Lang, Dennis Wei, Prasanna Sattigeri, Subhro Das, and David Sontag. Who should predict? exact algorithms for learning to defer to humans. In International Conference on Artificial Intelligence and Statistics, pp. 10520–10545. PMLR, 2023.
>
> [2] Rajeev Verma and Eric Nalisnick. Calibrated learning to defer with one-vs-all classifiers. In International Conference on Machine Learning, pp. 22184–22202. PMLR, 2022a.
>
> [3] Mohammad-Amin Charusaie, Hussein Mozannar, David A. Sontag, and Samira Samadi. Sample Efficient Learning of Predictors that Complement Humans. In Kamalika Chaudhuri, Stefanie Jegelka, Le Song, Csaba Szepesvári, Gang Niu, and Sivan Sabato (eds.), International Conference on Machine Learning, ICML 2022, 17-23 July 2022, Baltimore, Maryland, USA, volume 162 of Proceedings of Machine Learning Research, pp. 2972–3005. PMLR, 2022.
>
> [4] Rajeev Verma, Daniel Barrejón, and Eric Nalisnick. Learning to defer to multiple experts: Consistent surrogate losses, confidence calibration, and conformal ensembles. In International Conference on Artificial Intelligence and Statistics, pp. 11415–11434. PMLR, 2023.